# The endocannabinoid N-arachidonoyl dopamine is critical for hyperalgesia induced by chronic sleep disruption

Weihua Ding [1,11], Liuyue Yang[1,11], Eleanor Shi[1], Bowon Kim[1], Sarah Low[1], Kun Hu[2], Lei Gao[1], Ping Chen[3], Wei Ding[3], David Borsook[4], Andrew Luo[5], Jee Hyun Choi [6], Changning Wang[7], Oluwaseun Akeju [1], Jun Yang [7], Chongzhao Ran[7], Kristin L. Schreiber[8], Jianren Mao[1], Qian Chen[9,10], Guoping Feng [9]✉ & Shiqian Shen [1]✉

Chronic pain is highly prevalent and is linked to a broad range of comorbidities, including sleep disorders. Epidemiological and clinical evidence suggests that chronic sleep disruption (CSD) leads to heightened pain sensitivity, referred to as CSD-induced hyperalgesia. However, the underlying mechanisms are unclear. The thalamic reticular nucleus (TRN) has unique integrative functions in sensory processing, attention/arousal and sleep spindle generation. We report that the TRN played an important role in CSD-induced hyperalgesia in mice, through its projections to the ventroposterior region of the thalamus. Metabolomics revealed that the level of N-arachidonoyl dopamine (NADA), an endocannabinoid, was decreased in the TRN after CSD. Using a recently developed CB1 receptor (cannabinoid receptor 1) activity sensor with spatiotemporal resolution, CB1 receptor activity in the TRN was found to be decreased after CSD. Moreover, CSD-induced hyperalgesia was attenuated by local NADA administration to the TRN. Taken together, these results suggest that TRN NADA signaling is critical for CSD-induced hyperalgesia.

Chronic pain affects 11–40% of all adults[1] and is linked to a wide range of comorbidities, including anxiety, depression, and sleep disorders[2–4]. Notably, one-third of US adults report experiencing sleep disorders or inadequate sleep[5,6], and adults with chronic pain are more likely to experience sleep disorders[7]. Both epidemiological and clinical evidence suggests that chronic sleep disruption (CSD) promotes pain[8], which is consistent with the common life experience of generalized

hyperalgesia following a single night of sleep deprivation[9,10]. Despite empirical, clinical, and epidemiological evidence of the link between disturbed sleep and pain, the mechanisms through which CSD induces hyperalgesia are largely unknown[11].

In a seminal study, Alexandre et al. established a mouse model of CSD which involved minimal stress, and observed significantly increased pain sensitivity (referred to as 'CSD-induced

[1]Department of Anesthesia, Critical Care and Pain Medicine, Massachusetts General Hospital, Harvard Medical School, Boston, MA, USA. [2]Department of Pathology, Tuft University School of Medicine, Boston, MA, USA. [3]College of Science and Mathematics, University of Massachusetts Boston, Boston, MA, USA. [4]Department of Radiology, Massachusetts General Hospital, Harvard Medical School, Boston, MA, USA. [5]Summer Intern Program of the Department of Anesthesia, Critical Care and Pain Medicine, Massachusetts General Hospital, currently at Brandeis University, Boston, MA, USA. [6]Center for Neuroscience, Korea Institute of Science and Technology, Seoul, South Korea. [7]Martinos Center for Biomedical Imaging, Department of Radiology, Massachusetts General Hospital, Harvard Medical School, Boston, MA, USA. [8]Department of Anesthesiology, Perioperative and Pain Medicine, Brigham and Women's Hospital, Harvard Medical School, Boston, MA, USA. [9]McGovern Institute for Brain Research and Department of Brain and Cognitive Sciences, Massachusetts Institute of Technology, Cambridge, MA, USA. [10]Present address: Zhongshan Institute for Drug Discovery, Shanghai Institute of Materia Medica, Chinese Academy of Sciences, Shanghai, China. [11]These authors contributed equally: Weihua Ding, Liuyue Yang. ✉e-mail: fengg@mit.edu; sshen2@mgh.harvard.edu

hyperalgesia')[12].The neural substrates that mediate CSD-induced hyperalgesia have only begun to be unraveled, but implicate the nucleus accumbens, the hypothalamus-pituitary-adrenal axis, as well as adenosine and nitric oxide signaling[8,12–16]. The thalamic reticular nucleus (TRN) has unique integrative functions in sensory processing and nociception[17–19], attention/arousal and sleep spindle generation[8,20–25]. We hypothesized that the TRN may be involved in CSD-induced hyperalgesia. Using a mouse model of CSD, we report that the TRN played an important role in CSD-induced hyperalgesia, through its projections to the ventroposterior region (VP) of the thalamus. Chemogenetic activation of TRN neurons that projected to the VP of the thalamus attenuated CSD-induced hyperalgesia; chemogenetic inhibition of neurons in the VP receiving projections from the TRN also attenuated CSD-induced hyperalgesia. A metabolomic screening revealed that the level of *N*-arachidonoyl dopamine (NADA), an endocannabinoid, was decreased in the TRN after CSD. Moreover, endocannabinoid receptor 1 (CB1 receptor) activity decreased after CSD, and CSD-induced hyperalgesia was attenuated by local NADA administration to the TRN, demonstrating a critical role for NADA in CSD-induced hyperalgesia.

## Results

### The TRN is critical for CSD-induced hyperalgesia

We hypothesized that the TRN may play an important role in CSD-induced hyperalgesia because of its unique integrative functions in

sensory processing and nociception[17–19], attention/arousal and sleep spindle generation[8,20–25]. To test this hypothesis, we first adopted a CSD model as described[12]. Mice were deprived of sleep between 7 am and 1 pm daily for 5 consecutive days (Fig. 1a) by introducing novel objects when sleep attempts occurred (Fig. 1b). The degree of sleep deprivation was validated using a wireless electroencephalogram (EEG) recording as previously reported[26]. Immediately following CSD sessions, mice were allowed to sleep in their normal environment, during which wireless EEG was monitored between 1 and 3 pm. Results showed that the non-REM sleep duration of these mice was significantly increased after CSD sessions (Fig. 1c, Supplementary Fig. 1a), presumably due to compensatory mechanisms related to sleep homeostasis[27]. The increased non-REM sleep following CSD session was consistent with previous reports[12,28]. Notably, despite this compensation, total sleep time decreased significantly in mice that underwent CSD when compared with controls (Fig. 1d, Supplementary Fig. 1b), consistent with previously reported results[12]. Pain-related behavioral tests were conducted between 1–2 pm following CSD sessions, and mice displayed significantly decreased facial and hindpaw mechanical withdrawal thresholds and a decreased hindpaw thermal withdrawal latency, consistent with CSD-induced hyperalgesia in widespread areas of the body (Fig. 1e). In humans, sleep disruption leads to both high self-reported pain scores and increased pain sensitivity to mechanical and thermal stimuli, in widespread areas of the body[10,29–31].

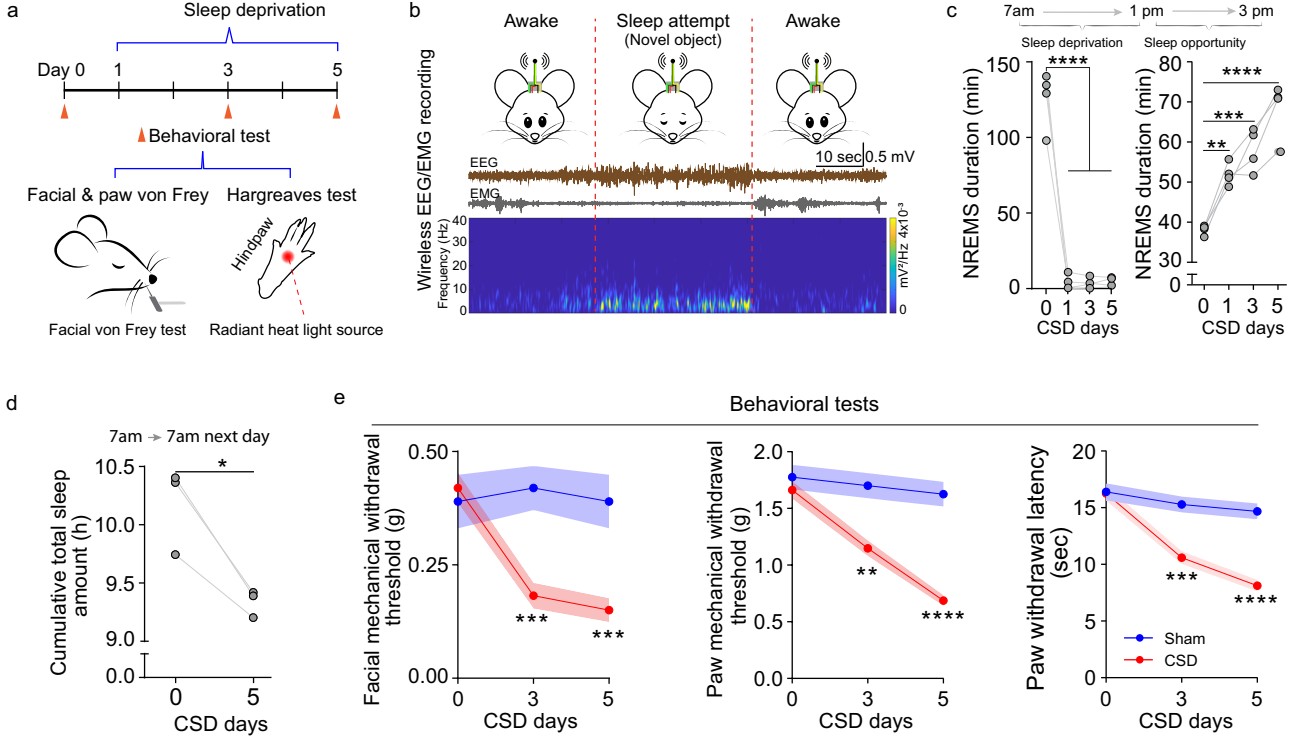

**Fig. 1 | CSD induces hyperalgesia. a** Experimental diagram of the sleep deprivation experiment and behavioral testing. **b** Sample EEG recording using wireless EEG recording. Brown line-EEG trace. Gray line-EMG trace. A representative sleep attempt on EEG and EMG traces during sleep deprivation. Lower panel: time-frequency representation of an EEG signal (spectrogram) showing increased slow activity (0.5–4 Hz) during a manually detected sleep attempt. **c** Duration of non-REM sleep during the sleep deprivation session (7 am–1 pm; *n* = 4 mice; Day 0 vs day 1, 3, 5 *p* < 0.0001) and during the representative sleep opportunity period (1 pm–3 pm, Day 0 vs day 1 *p* = 0.0033; Day 0 vs day 3 *p* = 0.0002; Day 0 vs day 5 *p* < 0.0001). One-way ANOVA indicated a statistically significant difference. Tukey's post hoc test. **\*\****p* < 0.01, **\*\*\*** *p* < 0.001, **\*\*\*\****p* < 0.0001. **d** Total cumulative sleep time over 24 h (7 am-7 am; *n* = 3 mice; Day 0 vs day 5 *p* = 0.03). Two-sided paired *t* test,

**\****p* < 0.05. **e** Facial mechanical withdrawal threshold (Sham vs CSD at day 3 *p* = 0.0003, day 5 *p* = 0.0003), hindpaw mechanical withdrawal threshold (Sham vs CSD at day 3 *p* = 0.0039, day 5 *p* < 0.0001), and hindpaw thermal withdrawal latency (Sham vs CSD at day 3 *p* = 0.0003, day 5 *p* < 0.0001) at the indicated time points. Testing was performed within 1 h after the CSD session. (Sham *n* = 8 mice, CSD *n* = 16). Two-way ANOVA indicated that the differences in behavioral parameters between the two groups were statistically significant. The Bonferroni post hoc test indicated that the differences at the indicated time points were significant. Data are presented as mean ± SEM, **\*\****p* < 0.01, **\*\*\****p* < 0.001, **\*\*\*\****p* < 0.0001. Source data are provided as a Source Data file. EEG electroencephalogram, EMG Electromyography, NREM non-rapid eye movement.

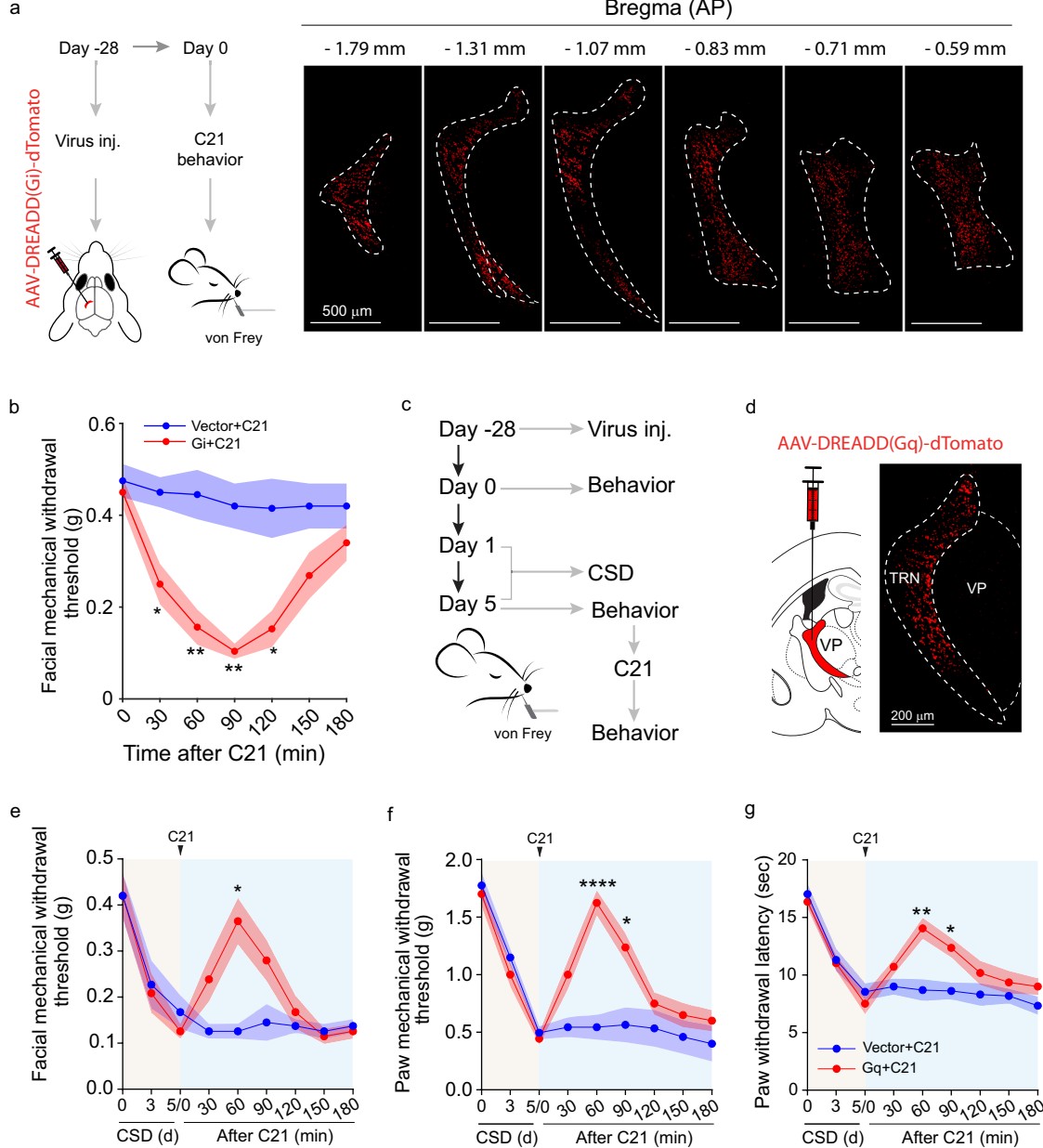

**Fig. 2 | TRN is critical for CSD-induced hyperalgesia. a, b** Chemogenetic inhibition of the TRN promoted nociceptive behavior. **a** Experimental diagram of targeted TRN inhibition using AAV8-hDlx-Gi GREADD-dTomato or AAV8-hDlx-dTomato as a control. The images were tangential brain slices from a representative mouse to demonstrate dTomato expression in the TRN. This experiment was repeated independently three times with similar results. **b** Facial mechanical withdrawal threshold at the indicated time points post-C21 administration. ($n = 8$ mice, Vector+C21 vs Gi+C21 at 30 min $p = 0.0206$, 60 min $p = 0.0055$, 90 min $p = 0.0015$, 120 min $p = 0.0342$). Two-way ANOVA indicated that the differences in behavioral parameters between the two groups were statistically significant. The Bonferroni post hoc test indicated that the differences were statistically significant at the indicated time points; *$p < 0.05$, **$p < 0.01$. **c–g** Chemogenetic activation of the TRN alleviated CSD-induced pain sensitivity. **c** Diagram of experimental design. AAV-hDlx-Gq GREADD-dTomato or the control AAV-hDlx-dTomato control was injected into the mouse TRN, and the mice were maintained for 4 weeks to allow virus expression. All mice were then subjected to CSD followed by C21 administration at 1 pm on the day of the last CSD session. **d** Representative image of TRN DREADDs expression and this experiment was repeated independently three times with similar results. The facial mechanical withdrawal threshold (Vector+C21 vs Gq +C21 at 60 min $p = 0.0166$) **e** hindpaw mechanical withdrawal threshold (Vector +C21 vs Gq+C21 at 60 min $p < 0.0001$, 90 min $p = 0.0378$) **f** and hindpaw thermal withdrawal latency (Vector+C21 vs Gq+C21 at 60 min $p = 0.0095$, 90 min $p = 0.0407$) **g** were examined at the indicated time points ($n = 8$ mice). Two-way ANOVA indicated that the differences in behavioral parameters between the two groups were statistically significant. The Bonferroni post hoc test indicated the differences were statistically significant at the indicated time points. Data are presented as mean ± SEM, *$p < 0.05$, **$p < 0.01$, ****$p < 0.0001$. Source data are provided as a Source Data file. DREADDs: designer receptors exclusively activated by designer drugs. VP ventral posterior nucleus of the thalamus.

To examine the potential role of the TRN in this phenomenon, TRN neurons were targeted using a viral targeting strategy[32]. TRN neurons mostly are GABAergic projection neurons[22,33,34]. Dlx promoter has been shown to efficiently target GABAergic neurons for functional manipulation[32]. To inhibit TRN neurons, an inhibitory type of designer receptors exclusively activated by designer drugs (DREADDs)[35], AAV-Dlx-Gi DREADDs-dTomato, was used (Fig. 2a, Supplementary Fig. 2a). To confirm the inhibitory effects of Gi DREADDs, a mixture of AAV-Dlx-Gi DREADDs-dTomato and AAV-Dlx-GCaMP6f was injected into the TRN of well-rested 'naive' mice. Fiber photometry imaging of the TRN

showed that Gi DREADDS led to decreased calcium activities (Supplementary Fig. 2b, c), consistent with decreased TRN activities. When TRN was inhibited using AAV-Dlx-Gi DREADDs, there were significantly decreased facial and hindpaw mechanical withdrawal thresholds and hindpaw thermal withdrawal latency (Fig. 2b, Supplementary Fig. 2d, e), suggesting that TRN inhibition led to hyperalgesia. Additionally, we tested if activation of TRN neurons could alleviate CSD-induced hyperalgesia. For this, an excitatory DREADDs AAV-Dlx-Gq DREADDs was injected into the TRN (Fig. 2c, d). These mice subsequently underwent CSD. Chemogenetic activation of TRN neurons using Gq DREADDs significantly increased the facial and hindpaw mechanical withdrawal thresholds and hindpaw thermal withdrawal latency (Fig. 2e–g), suggesting that TRN activation attenuated CSD-induced hyperalgesia. In addition to chemogenetic approaches, we also used optogenetic approaches to inhibit and active TRN as described[36,37].

Similar to the experiments using chemogenetics, optogenetic TRN inhibition led to hyperalgesia in 'naive' mice (Supplementary Fig. 3a–d), whereas optogenetic TRN activation led to attenuation of CSD-induced hyperalgesia (Supplementary Fig. 3e–h). Thus, using both chemogenetic and optogenetic manipulation to activate or inhibit TRN, our results implicate a role for the TRN in CSD-induced hyperalgesia.

## The TRN-to-VP circuit is implicated in CSD-induced hyperalgesia

To directly assess TRN neuronal dynamics in CSD-induced hyperalgesia, the calcium sensor GCaMP6f was expressed in the TRN by AAV-hDlx-GCaMP6f (Fig. 3a). Fiber photometry was performed to image calcium dynamics prior to and after five sessions of sleep deprivation (Fig. 3a, Supplementary Fig. 4a). TRN neuronal activity was captured with calcium dynamics and area under the curve of calcium traces was

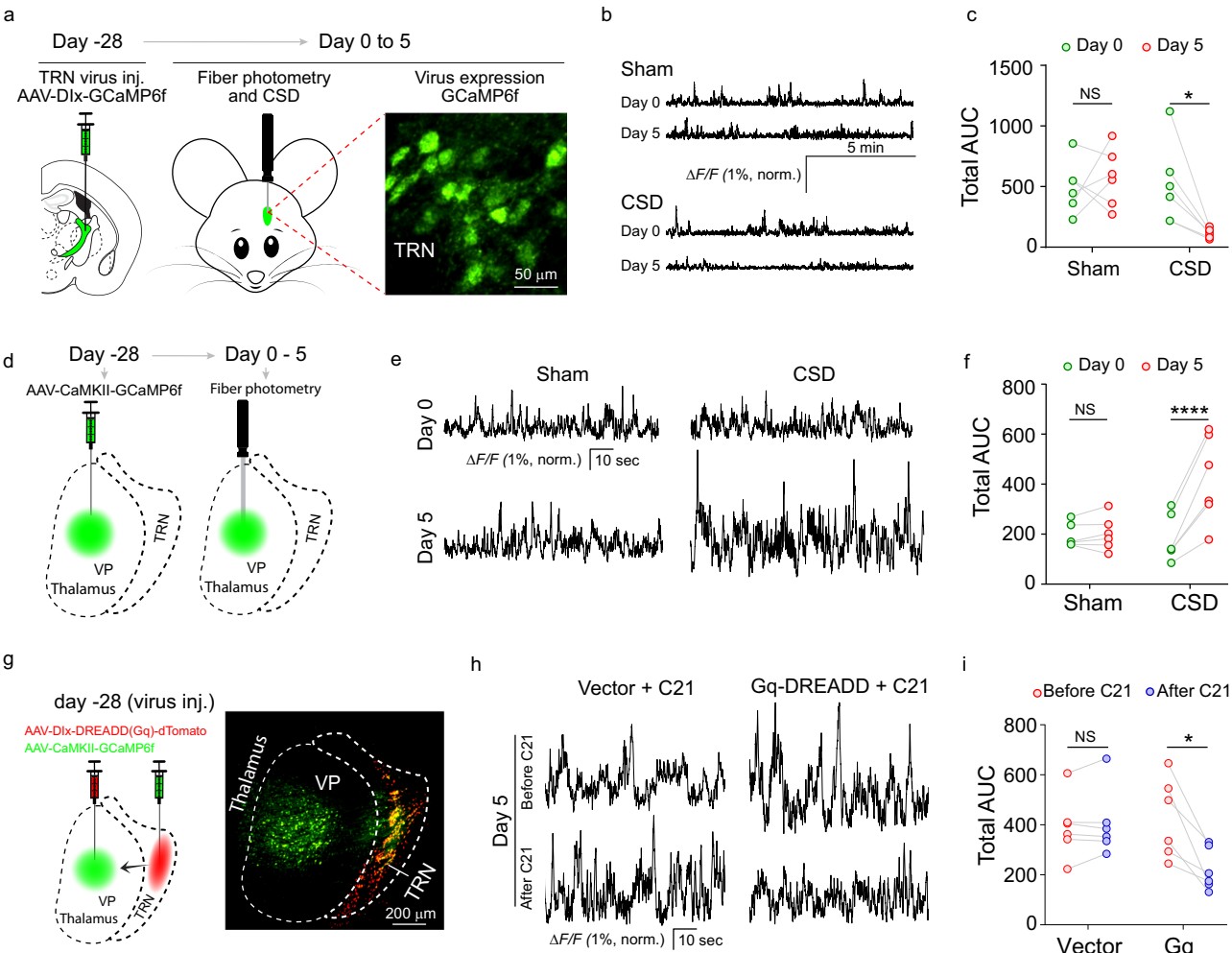

**Fig. 3 | Decreased neural activity in the TRN after CSD. a–c** Neural dynamics in the TRN before and after CSD. **a** Experimental diagram of the experiment used to study neural dynamics in the TRN. Mice in the sham and CSD groups (*n* = 6 mice) were injected with AAV-hDlx-GCaMP6f into the TRN and maintained for 4 weeks to allow virus expression. Fiber photometry recording was performed on day 0 and day 5. The fluorescence picture shows virus expression in the TRN. **b** Sample traces of calcium signals in the TRN. **c** Statistical analysis of the area under the curve for calcium traces in the sham and CSD groups (*n* = 6 mice, Day 0 vs day 5 CSD *p* = 0.0147); \**p* < 0.05, two-sided paired *t* test. **d–f** Calcium dynamics in the VP. **d** Diagram of the experiment used to study neural dynamics in the VP. Mice in the sham and CSD groups (*n* = 6 mice) were injected with AAV-CaMKII-GCaMP6f into the VP and maintained for 4 weeks to allow virus expression. Fiber photometry recording was performed on day 0 and day 5. **e** Sample traces of calcium dynamics in the VP. **f** Statistical analysis of the area under the curve of calcium traces in the sham and CSD groups (*n* = 6 mice, Day 0 vs day 5 CSD *p* < 0.0001). Two-sided paired *t* test. **g–i** Chemogenetic activation of the TRN attenuated CSD-induced VP hyperactivity. **g** Experimental diagram of virus injection to the TRN and VP. AAV-hDlx-Gq DREADD-dTomato (*n* = 6 mice) or AAV-hDlx-dTomato vector control (*n* = 6 mice) was injected into the TRN. AAV-CaMKII-GCaMP6f was injected into the VP of all mice. Four weeks were allowed for virus expression. Fiber photometry was performed to measure neural activity in the VP. A representative image of virus expression is shown. **h** Sample traces of neural activity in the VP after 5 sessions of CSD (before and 30 min after C21 administration). **i** Statistical analysis of the area under the curve of calcium traces before and after C21 administration (*n* = 6 mice, Gq before vs after C21 *p* = 0.0312). Two-sided paired *t* test. \* *p* < 0.05. Source data are provided as a Source Data file. AUC area under the curve.

used for quantification (Fig. 3b, c). The results showed that TRN neuron activity decreased significantly after CSD when compared with pre-CSD baseline (Fig. 3c), further supporting TRN's involvement in CSD-induced hyperalgesia.

The TRN is known for its topographic projections to various thalamic subnuclei. For example, the somatosensory TRN projects to the ventrobasal complex of the thalamus, including the VP[38–41]. To confirm that the TRN projects to the VP, AAV1-hSyn-eGFP (nls), which expresses GFP with a nuclear localization signal[42], was used. AAV1 has been shown to mediate anterograde transsynaptic tagging which enables probing of neural circuitry[43,44]. Projections from the TRN to the VP would therefore lead to the expression of nuclear GFP in VP neurons. Our results indeed showed that injection of AAV1-hSyn-eGFP (nls) led to robust expression of GFP in the VP (Supplementary Fig. 4b). When these neurons were co-stained for glutamate, nearly all GFP-positive neurons were also positive for glutamate (Supplementary Fig. 4c–e). TRN is enriched for PV expressing neurons, which are absent in mouse thalamus[25]. Using a previously established strategy[25], AAV1-hSyn-DIO-GFP was injected into the TRN of PV-Cre mice. This injection led to robust expression of GFP in the TRN and the VP region, confirming TRN to VP projections (Supplementary Fig. 5a, b). To corroborate these findings, AAVretro-hSyn-eGFP (nls) was injected into the VP (Supplementary Fig. 6a), and robust GFP expression was found in the TRN region, consistent with the TRN to VP projections. When co-stained for GABA, nearly all retrogradely targeted GFP-positive neurons were also positive for GABA (Supplementary Fig. 6b). As a validation of the AAVretro targeting strategy, we also examined the spinal trigeminal nucleus, a region known to relay to the VP, and found that GFP-positive cells were present in this region (Supplementary Fig. 6c, d). As such, both anterograde and retrograde tracing confirmed that the TRN projects to the VP.

After CSD, c-Fos expression in the VP region increased, suggesting activation of this brain region in CSD (Supplementary Fig. 7a–c). To directly assess VP neural activities, GCaMP6f was expressed in the VP using AAV8-CaMKII-GCaMP6f for fiber photometry examination of calcium dynamics. The calcium dynamics in the VP were increased, consistent with an increase of VP neuronal activity (Fig. 3d–f). We postulated that the observed increase of VP neuronal activity after CSD was due to decreased TRN neuronal activity. If so, increasing TRN neuronal activity would dampen VP hyperactivity after CSD. To test this, AAV8-hDlx-Gq DREADDs and AAV9-CaMKII-GCaMP6f were injected into the TRN and VP, for activating TRN neurons and imaging VP neuronal dynamics, respectively (Fig. 3g). In mice subjected to CSD, DREADDs-mediated activation of the TRN was accompanied by a decrease in neural dynamics in the VP (Fig. 3h, i), confirming a functional impact of the TRN on VP neural activity. The functional impact of the TRN on VP neural activity was also confirmed using optogenetic TRN activation, which alleviated CSD-induced VP hyperactivities (Supplementary Fig. 8a, b). To further study the behavioral implication of the functional impact, AAV1-hSyn-eGFP-Cre was injected into the TRN, and Cre-dependent AAV8-DIO-Gi DREADDs was injected into the VP (Fig. 4a, b). Due to the anterograde transsynaptic transmission of AAV1, this set up enabled Gi DREADDs expression in VP neurons that were receiving inputs from the TRN. In mice subjected to CSD, chemogenetic inhibition of VP neurons receiving inputs from the TRN significantly increased the withdrawal parameters, including the facial (Fig. 4c) and hindpaw (Fig. 4d) mechanical withdrawal thresholds and hindpaw thermal withdrawal latency (Fig. 4e). These results suggest that the TRN to VP projection is functionally important for CSD-induced hyperalgesia. To confirm this, a parallel set of experiments were performed. AAVretro-hSyn-eGFP-Cre was injected into the VP, and Cre-dependent AAV8-DIO-Gq DREADDs were injected into the TRN in mice subjected to CSD. TRN neurons that were retrogradely targeted from the VP expressed the Gq DREADDs (Fig. 4f, g). Chemogenetic activation of these neurons significantly increased the

nociceptive thresholds, including the facial (Fig. 4h) and hindpaw (Fig. 4i) mechanical withdrawal thresholds and hindpaw thermal withdrawal latency (Fig. 4j). As such, both anterograde and retrograde targeting experiments suggest that TRN projections to the VP likely play an important role in CSD-induced hyperalgesia.

Besides VP neural activities in awake and resting state, we also measured its activities in response to mechanical stimulation. Von Frey (mechanical stimulation) led to significantly higher neural activities in mice that underwent CSD than in well-rested 'naive' mice (Supplementary Fig. 9a, b). More importantly, the VP hyperreactivities were dampened by chemogenetic activation of the TRN (Supplementary Fig. 9c, d).

## The level of N-arachidonoyl dopamine (NADA), an endocannabinoid, is decreased in the TRN after CSD

A targeted metabolomic screening was performed to evaluate the metabolomic changes in the TRN induced by CSD. Mice that underwent CSD were compared with control mice for their metabolites, using a fold change of two and p value of 0.05 as cutoffs. N-arachidonoyl dopamine (NADA) levels were decreased in the TRN in the CSD group compared with the control group (Fig. 5a, b, Supplementary Fig. 10a, Supplementary data 1). Levels of glycerol-3-phosphate and reduced glutathione in the TRN were increased in the CSD group. In the thalamus and the primary somatosensory cortex, adenosine monophosphate, adenine, and inosine monophosphate levels were increased in the CSD group (Supplementary Fig. 10b, c), consistent with the known role of nucleotides, including purines, in nociception and pain[45,46]. NADA is an endocannabinoid that acts as an agonist for both cannabinoid receptor 1 (CB1 receptor) and transient receptor potential V1 (TRPV1)[47–49]. Immunofluorescence staining for CB1 and TRPV1 showed that TRPV1 expression was negligible, whereas CB1 receptor was abundantly expressed in the TRN (Fig. 5c, d). We hypothesized that decreased levels of NADA in the TRN would lead to decreased CB1 receptor activity after CSD. A recently developed sensor for CB1 receptor activity with spatiotemporal resolution, GRAB-eCB2.0[50], was used to assess TRN CB1 receptor activity. Specifically, AAV8-hSyn-GRAB-eCB2.0 was expressed in the TRN for fiber photometry examination (Fig. 5e, Supplementary Fig. 10d–f). CB1 receptor agonist ACEA[51] increased TRN GRAB-eCB2.0 activities in a control experiment (Supplementary Fig. 11a, b), validating its utility in reporting CB1 receptor activities. CSD led to a significant decrease in CB1 receptor activity as quantified by area under the curve of CB1 receptor activity traces (Fig. 5f, g). The decreased TRN CB1 receptor activity was consistent with the decreased levels of NADA after CSD.

## NADA in the TRN is critical for CSD-induced hyperalgesia

CSD led to decreased NADA levels and decreased CB1 receptor activities in the TRN (Fig. 5), raising the possibility that decreased NADA levels in the TRN might be linked to CSD-induced hyperalgesia. To test this, the TRN was cannulated for NADA administration (Fig. 6a, Supplementary Fig. 12a). In mice subjected to CSD, administration of NADA led to decreased nociceptive sensitivity, in the form of higher hindpaw mechanical withdrawal threshold (Fig. 6b), higher hindpaw thermal withdrawal latency (Fig. 6c), and higher facial mechanical withdrawal threshold (Fig. 6d), when compared with administration of artificial cerebral spinal fluid (ACSF) as control. These results suggest that exogenous supplementation of NADA to the TRN could attenuate CSD-induced hyperalgesia. When a CB1 receptor antagonist was coadministered, the inhibition of CSD-induced hyperalgesia by NADA was largely abrogated (Supplementary Fig. 13a–d), supporting a critical role of NADA-CB1 receptor in CSD-induced hyperalgesia.

We showed that CSD-induced hyperalgesia was accompanied by increased VP neural dynamics, due to decreased TRN activities (Fig. 3b, c). To examine the neural dynamics associated with NADA administration, the TRN was cannulated for NADA administration,

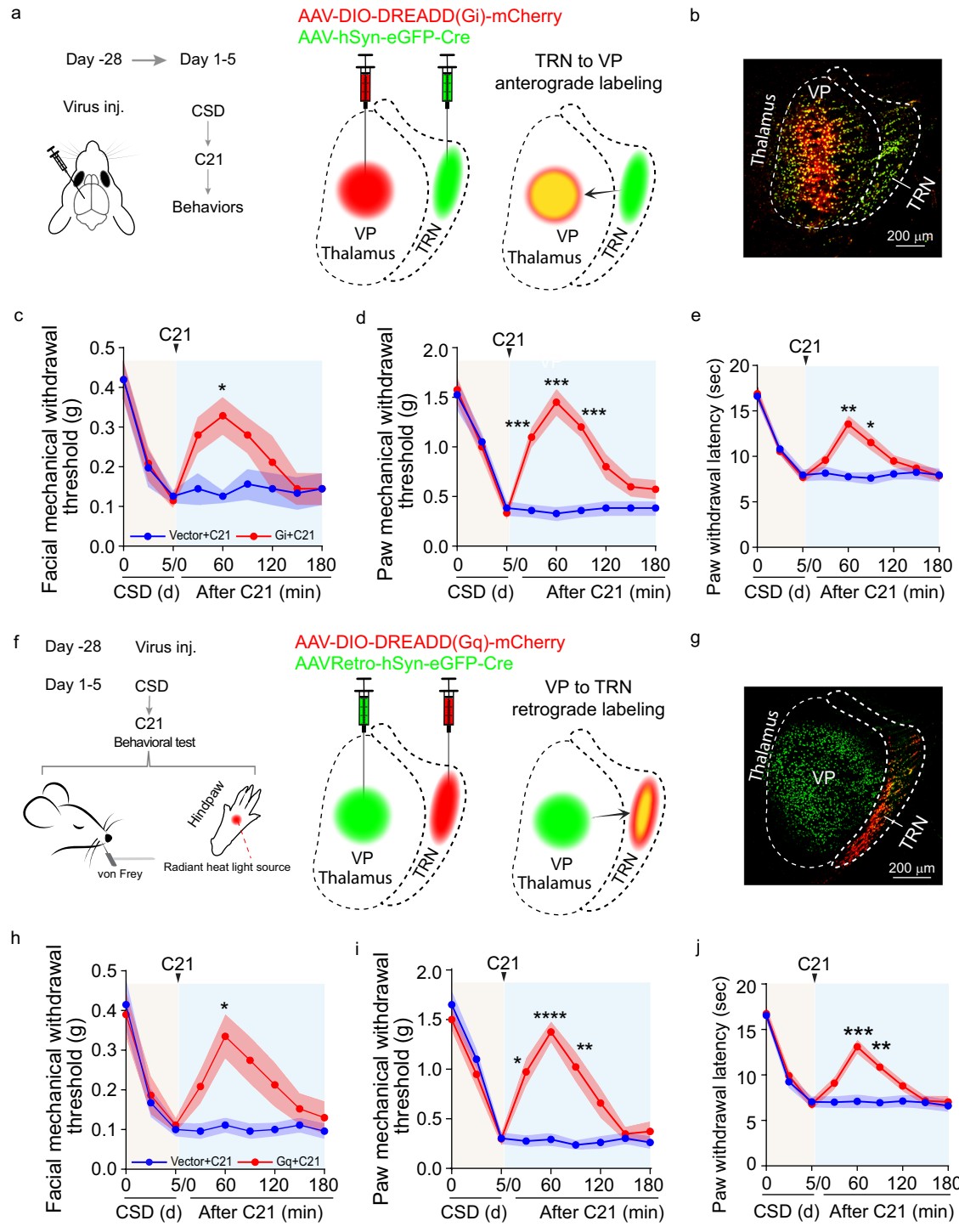

and the VP was injected with AAV9-CaMKII-CCaMP6f (Fig. 6e, Supplementary Fig. 14a, b) for neural dynamics assessment with fiber photometry (Fig. 6f, Supplementary Fig. 14c). In mice subjected to CSD, ACSF did not significantly change VP neural activity, whereas NADA significantly dampened VP hyperactivity induced by CSD (Fig. 6g, h). The dampened VP neural dynamics were consistent with NADA's ability to attenuate CSD-induced hyperalgesia.

## Discussion

Patients with chronic pain often report comorbid sleep disorders[3,8,11]. Sleep disruption itself induces nociception and exaggerates pain perception. As such, CSD-induced hyperalgesia adds to the challenges of effective pain treatment. In this report, we adopted a mouse model of CSD and found that the TRN was important for CSD-induced hyperalgesia, through its projections to the VP—a brain region relays somatosensory information. More interestingly, metabolomic experiments revealed that the endocannabinoid NADA was critical for CSD-induced hyperalgesia, as TRN administration of NADA attenuated CSD-induced hyperalgesia and dampened the exaggerated VP neural activities in CSD, the effect of which could be blocked using a CB1 antagonist. These findings provide mechanistic insights into the neuronal circuitry underlying CSD-induced hyperalgesia, and implicate endocannabinoids as potential mechanistic targets for future study.

NADA is an N-acryl amino acid, and these amino acids were recently recognized to be bioactive lipids linked to metabolism and nociception[52]. Previously, NADA has been shown to exhibit systematic

**Fig. 4 | The TRN to VP projection is critical for CSD-induced hyperalgesia.**
**a–e** Chemogenetic inhibition of VP neurons receiving inputs from the TRN alleviated CSD-induced hyperalgesia. AAV1-hSyn-eGFP-Cre was injected into the TRN. AAV-hSyn-DIO-Gi DREADD-mCherry (*n* = 8 mice) or AAV-hSyn-DIO-mCherry vector control (*n* = 8 mice) was injected into the VP. All mice were subjected to CSD sessions, and C21 was administered at the end of the 5th CSD session. Behavioral testing was performed at the indicated time points. **a** Diagram of the virus targeting strategy. **b** A representative virus expression picture for the TRN and VP regions. Green: GFP; red: mCherry. The facial mechanical withdrawal threshold (vector+C21 vs Gi+C21 at 60 min *p* = 0.0287) **c** hindpaw mechanical withdrawal threshold (vector+C21 vs Gi+C21 at 30 min *p* = 0.0004, 60 min *p* = 0.0002, 90 min *p* = 0.0002) **d** and hindpaw thermal withdrawal latency (vector+C21 vs Gi+C21 at 60 min *p* = 0.002, 90 min *p* = 0.0273) **e** of mice that received Gi DREADD and vector control were examined at the indicated time points. **f–j** Chemogenetic activation of the TRN neuron that were retrogradely targeted from the VP alleviated CSD-induced pain sensitivity. **f** AAVretro-hSyn-eGFP-Cre was injected into the VP.

AAV-hSyn-DIO-Gq DREADD-mCherry (*n* = 8) or the control AAV-hSyn-DIO-mCherry vector (*n* = 8 mice) was injected into the TRN. **g** A representative virus expression picture for the TRN and VP regions. Green: GFP; red: mCherry. All mice were subjected to CSD sessions, and C21 was administered at the end of the 5th CSD session. Behavioral testing was performed at the indicated time points. The facial mechanical withdrawal threshold (vector+C21 vs Gq+C21 at 60 min *p* = 0.0448) **h** hindpaw mechanical withdrawal threshold (vector+C21 vs Gq+C21 at 30 min *p* = 0.0121, 60 min *p* < 0.0001, 90 min *p* = 0.0095) **i** and hindpaw thermal withdrawal latency (vector+C21 vs Gq+C21 at 60 min *p* = 0.0009, 90 min *p* = 0.0046) **j** of mice that received Gq DREADD and vector control were shown. Two-way ANOVA followed by Bonferroni's multiple comparisons test was used. Data are presented as mean ± SEM, **p* < 0.05, ***p* < 0.01, ****p* < 0.001, *****p* < 0.0001 at the indicated time points. Experiments of **b** and **g** were repeated independently three times with similar results. Source data are provided as a Source Data file. VP ventroposterior thalamus.

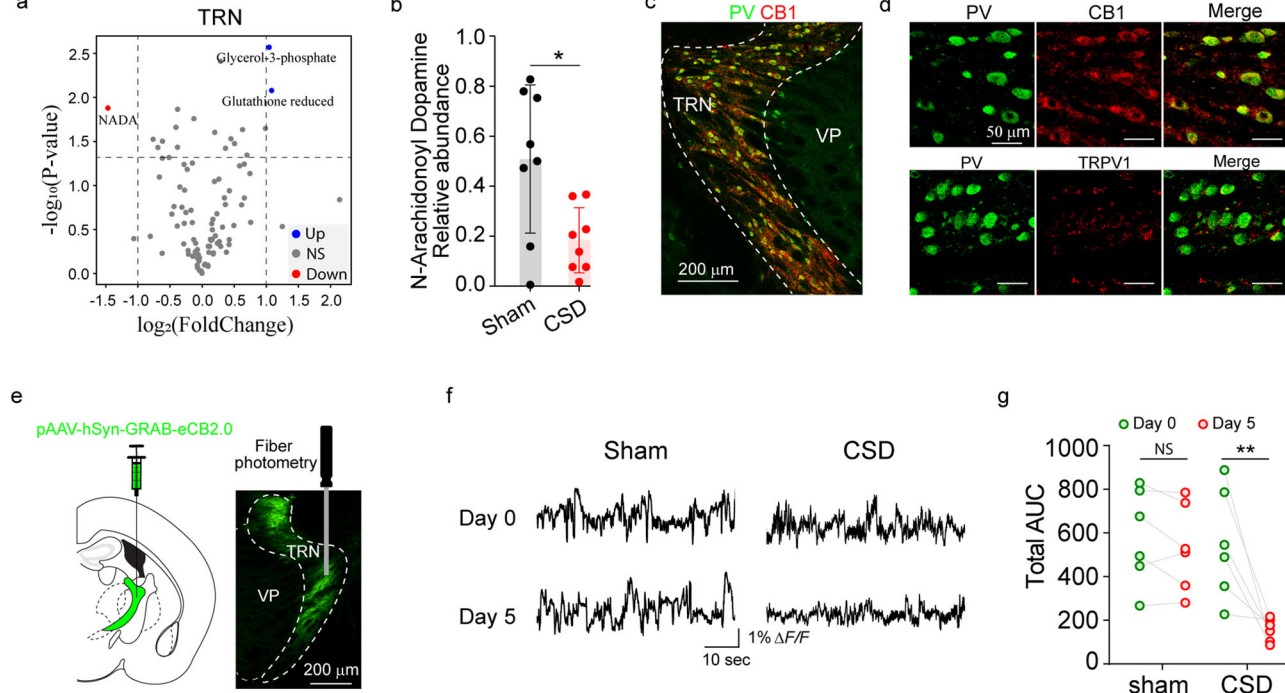

**Fig. 5 | The level of NADA is decreased in the TRN after CSD. a–b** Targeted metabolomics analysis of the TRN. Mice in the CSD and sham groups (*n* = 8 mice). **a** Volcano plot of metabolites. Differential abundant metabolites were identified using a fold change of 2 and *p* value (*t* test) <0.05 as cutoffs. **b** Relative abundance of NADA (*n* = 8 mice, Sham vs CSD *p* = 0.0132). **p* < 0.05, two-sided *t* test. **c, d** Expression of the CB1 receptor in the TRN. **c** The picture is representative of six independently stained samples. **d** TRPV1 expression was negligible in the TRN. Anti-PV, anti-TRPV1, and anti-CB1 receptor antibodies were used. **e–g** Decreased CB1 receptor activity after CSD. AAV-hSyn-GRAB-eCB2.0 was injected into the TRN

(*n* = 6 mice). Four weeks were allowed for virus expression. Mice were then subjected to CSD. CB1 receptor activity was examined on days 0 and 5. **e** Diagram of the virus expression and fiber photometry experiments. The image was obtained after staining with an anti-GFP antibody. **f** Representative traces of CB1 receptor activity. **g** Area under the curve of TRN CB1 receptor activity traces (*n* = 6 mice, Day 0 vs day 5 CSD *p* = 0.0014). ***p* < 0.01, two-sided paired *t* test; NS: *p* > 0.05. Source data are provided as a Source Data file. NADA N-arachidonoyl dopamine, PV parvalbumin, CB1 cannabinoid receptor type 1, TRPV1 transient receptor potential vanilloid 1.

anti-inflammatory effects which was likely mediated by neuronal TRPV1[53]. The TRN displayed negligible TRPV1 expression, consistent with highly restricted expression of TRPV1 in the adult central nervous system[54,55]. As such, our observed effects of NADA were unlikely mediated by TRPV1. The pharmacological effects of NADA have been examined which showed both pronociceptive and antinociceptive properties[48,56,57]. However, the physiological roles of NADA are largely unknown. Our results suggest that NADA is physiologically important and that CSD leads to decreased NADA levels which underlies hyperalgesia.

Our findings that TRN is critical for CSD-induced hyperalgesia is consistent with TRN's unique role in both sensory processing and

sleep[8,20–25]. In our report, NADA was decreased after CSD, and this decrease was only found in the TRN, but not in the somatosensory cortex or thalamus, unraveling brain region-specific changes induced by CSD. Mechanistically, this role of TRN is likely mediated by its projections to the VP. Moreover, both decreased NADA levels and dampened CB1 receptor activities in the TRN are consistent with a critical role of TRN NADA signaling in CSD-induced hyperalgesia.

CB1 receptors located at the presynaptic terminals negatively regulate neuronal activities through retrograde signaling[58,59], which implies that decreased NADA levels would enhance TRN activities. This might seem at odds with our observation of decreased NADA and TRN activities. It is plausible that non-retrograde endocannabinoid

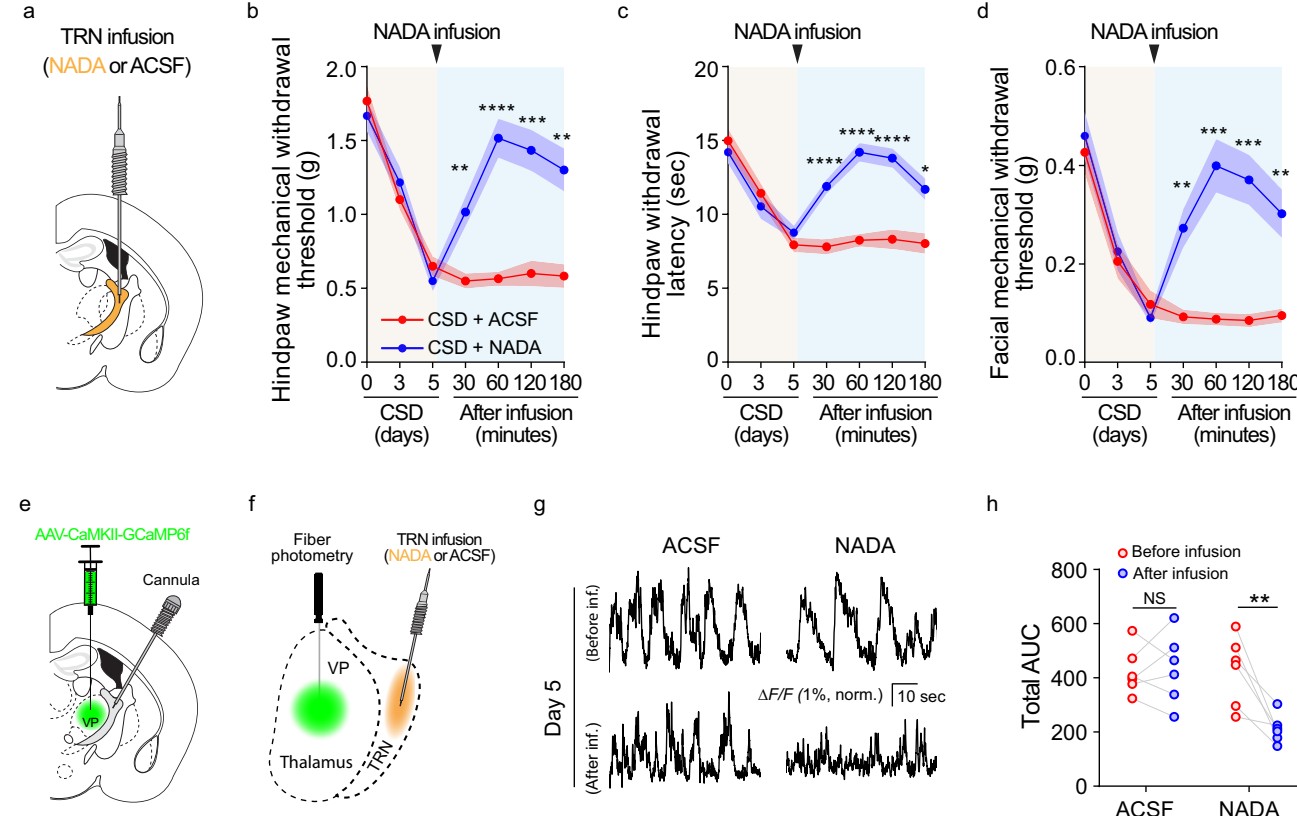

**Fig. 6 | NADA in the TRN is critical for CSD-induced hyperalgesia. a–d** Administration of NADA into the TRN alleviated CSD-induced hyperalgesia. The TRN was cannulated for NADA or ACSF administration. **b** The hindpaw mechanical withdrawal threshold (CSD + ACSF vs CSD + NADA at 30 min $p = 0.0049$, 60 min $p < 0.0001$, 120 min $p = 0.0005$, 180 min $p = 0.0031$), **c** hindpaw thermal withdrawal latency (CSD + ACSF vs CSD + NADA at 30, 60, 120 min $p < 0.0001$, 180 min $p = 0.0104$), and **d** facial mechanical withdrawal threshold (CSD + ACSF vs CSD + NADA at 30 min $p = 0.0050$, 60 min $p = 0.0007$, 120 min $p = 0.0009$, 180 min $p = 0.0096$) were determined at the indicated time points ($n = 8$ mice). Two-way ANOVA followed by Bonferroni's multiple comparisons test indicated that the differences between the two groups were statistically significant. Data are presented as mean ± SEM, $*p < 0.05$, $** p < 0.01$, $***p < 0.001$, $****p < 0.0001$. **e–h** Administration of NADA into the TRN dampened CSD-induced VP hyperactivity. The TRN was cannulated for NADA ($n = 6$ mice) or ACSF ($n = 6$ mice) administration, and AAV-CaMKII-GCaMP6f was injected into the VP. Four weeks were allowed for virus expression. Fiber photometry was performed on animals that received ACSF and NADA. **e–f** Diagram of TRN cannulation and fiber photometry of the VP. **g** Sample traces of VP calcium signals on day 5 of CSD before and after NADA administration. **h** Area under the curve of calcium traces in the VP ($n = 6$ mice, before vs after NADA infusion $p = 0.005$). $**p < 0.01$, two-sided paired $t$ test. Source data are provided as a Source Data file. NADA: $N$ arachidonoyl dopamine; ACSF artificial cerebral spinal fluid.

signaling pathways were implicated in our observations. For example, CB1 receptor has been shown to localize in mitochondrial membrane[60–62], which plays fundamental roles in neuronal energy metabolism and host behaviors. For example, CB1 receptor has been shown to protect mitochondria function and alleviate oxidative stress[63–65]. In line with these reports, our TRN metabolomics data showed that upregulated metabolites after CSD include glycerol-3 phosphate and glutathione (Fig. 5a), both of which are key molecules implicated in energy metabolism and oxidative stress[66,67]. Sleep loss has been shown to increase oxidative stress in both flies and mice[68].

The endocannabinoids are pleiotropic signaling molecules implicated in many neurological disorders, including Parkinson disease, Alzheimer disease, multiple sclerosis, epilepsy, etc.[69]. Our results provide insights into NADA, an endocannabinoid, in CSD-induced hyperalgesia and potential therapeutic strategies for heightened pain perception in individuals with chronic sleep disturbance.

## Methods
### Animal
All experimental procedures and animal use were reviewed and approved by the Institutional Animal Care and Use Committee (IACUC) of Massachusetts General Hospital. All performed experiments were conforming to the guidelines established by National Institutes of Health and the International Association for the Study of Pain. Adult male and female C57/BL6 and PV-Cre knockin (Jax 008069) mice (20–25 grams) were purchased from the Jackson Laboratory (ME). All experimental equipment used between different sexes testing were cleaned and disinfected. Mice were housed in the same condition with $n = 4$ per cage with males and females separated. Animal room was climate-controlled with 12-hour light–dark cycles (lights on at 07:00 am, lights off at 07:00 pm) at a stable temperature of $23 \pm 1 °C$ and a consistent humidity of $50 \pm 5\%$, food and water were available ad libitum. CB1 receptor antagonist SR141716A (Tocris, Catlog 0923) was injected at 10 mg/kg intraperitoneally. CB1 receptor agonist arachidonyl-2′-chloroethylamide (Santa cruz, CAS 220556-69-4) was administered at 7.5 mg/kg intraperitoneally.

### Chronic sleep deprivation
Chronic sleep deprivation procedure was performed as previously described[12] with minor modifications. Specifically, all mice underwent five consecutive days of sleep deprivation sessions from 7:00 am to 1:00 pm. Baseline behaviors were tested one day prior to sleep deprivation, repeated behavioral tests were conducted between 1:00 pm to 2 pm at day 3 and day 5 of sleep deprivation sessions respectively. At the onset of sleep deprivation session, all the mice were kept awake by providing new home cage with new nesting beddings. Mice cages were

placed in the same room with minimum external disturbance during each sleep deprivation session. When necessary, slightly tapping the cage or providing novel objects in the animal cages. The quality of sleep deprivation was validated using wireless EEG and EMG to assure that mice were staying fully awake during each session. In order to promote activity with wakefulness, unalike objects with dissimilar shape or texture were used for each session. Mice were immediately allowed to sleep at day 1, 2 and 4 for the next 17, and for the next 16 h at day 3 and 5 after behavioral test until the next sleep deprivation session. For Fig. 1e CSD-induced hyperalgesia, male and female mice at equal numbers were used. For other experiments, male mice were used.

## EEG and EMG
Electrodes implantation for EEG and EMG was performed 3 weeks prior to sleep deprivation, mouse was anesthetized by isoflurane inhalation (3% for induction and 1.2−1.5% for maintenance). Mouse was placed on a stereotaxic frame and the dorsal fur of mouse head was removed using a micro sessors. Four handmade electrodes made of 0.8 mm diameter micro screws (custom order, Asia Bolt, South Korea) were fixed into the holes drilled in the mouse skull, caring not to damage the cortical area by inserting screws too deep. Two screw electrodes were implanted on the frontal area (AP 1.9 mm, ML 1 mm) and posterior area above the hippocampus (AP 2.2 mm, ML 2.2 mm). Two ground and reference screw electrodes were fixed into the Interparietal bone above the cerebellum. For EMG recording, a tungsten wire electrode (PFA-Coated Tungsten Wire, A-M systems, USA) inserted and hooked in neck muscles near occipital bone. All electrodes were secured and attached to the skull permanently with dental cement (C&B-Metabond, #171032. Parkell. Edgewood, NY USA). After surgery, antibiotics were applied, and mice had a recovery period for 3 weeks. CBRAIN device (2.6 g headstage, designed by Choi et al.[26] is used for wireless EEG signal monitoring and acquisition. Animals were habituated to the CBRAIN at least two days before experiment. EEG and EMG signals were continuously transmitted to a computer, monitored in real-time and recorded at 256 Hz in group-housed freely moving animals, starting at light-on time. Because of battery limitations, animals were perturbed a few seconds in every 6 h of recording while changing the fully charged CBRAIN devices.

EEG signals were viewed and analyzed off-line using the MATLAB (Mathworks Inc., USA). The raw text files loaded and reconstructed the period of missed data majorly caused by transmit errors of Bluetooth. The counter signal generated in CBRAIN was recorded together with physiological signals and used for identifying the missed data points. For preprocessing, signals were smoothed at every 2-s data points (512 points) to detrend any slow fluctuation and noises. The raw signals of EEG and EMG were plotted together for sleep scoring. Sleep stages were manually scored for every second. Time-frequency analyses were processed using the Built-in MATLAB function (spectrogram), which applied FFT to 1-s data point with 0.1-s moving window.

## Mechanical withdrawal threshold
All mice were acclimated to the behavioral examiner and testing environment 30−40 min for 3 consecutive days prior to baseline behavioral testing. Behavior was tested for the contralateral side of TRN manipulations. For facial mechanical withdrawal threshold test, each mouse was placed in a customized enclosure (6 × 6 × 6 m) with the top, bottom and four walls made of metal mesh. For hindpaw mechanical withdrawal threshold test, mice were individually placed in standard plexiglass enclosures with mesh grid floor. After 20−30 min of acclimation, mechanical sensitivity was measured using a graded series of von Frey filaments by inserting through mesh walls for facial sensitivity testing and mesh floor for plantar sensitivity testing. Starting with the filament generating the lowest force, the stimuli were applied within the trigeminal nerve or sciatic nerve territory of the animal for 1 s at 10 s

intervals. Brisk withdraw of the head/hindpaw, escaping or biting upon the stimuli was defined as positive withdrawal response and at least three positive responses out of 5 applications was defined as withdrawal threshold, and 2 g was defined as the final threshold if mice exhibited negative responses to all stimuli applications.

## Hindpaw withdrawal latency
Thermal latency to heat stimuli was assessed using Hargreaves procedure[70] with minor modifications. Prior to the testing, all mice were individually placed in standard Plexiglas enclosures on a pre-heated glass platform (28−29 °C) 20−30 min for acclimatization daily for 3 consecutive days. Emission of radiant heat source under the glass and perpendicularly focused to the midplantar surface. Paw withdrawal latency was defined as the duration (seconds) from the initiation of heat exposure to the hindpaw withdrawal, and 20 s was set as a cutoff time to avoid hindpaw injury.

## Virus injection and optical fiber implantation
Mice were anesthetized with oxygen mixed with isoflurane (3% for induction and 1.5% for maintenance) while monitoring respiratory rate. Eye lubricant was applied to moisten the eyes of mice. Ketorolac tromethamine (Althenex, Schaumburg, IL USA) was administrated (5 mg/kg) intraperitoneally every 12 h for 2 consecutive days to minimize postoperative pain. The head fur between outer canthus and concha level was shaved, then the mouse was mounted onto a stereotactic frame with ear bars. The front teeth were placed over the incisor bar. Mouse snout was then covered by a metal anesthetic mask delivering anesthetic gases. The skin was prepared with Povidone-Iodine solution (Aplicare, INC., Neriden, CT USA) followed by 70% alcohol swab (BD, Franklin Lakes, USA). After application of Lidocaine (0.1−0.2 ml, 1%), a skin flap overlying the dorsal skull was removed. Periosteum of the parietal skull and connective tissues was cleaned using cotton swabs dipped in saline.

For TRN injection, 10−15 nl of AAV9-hDlx-Gi DREADD-dTomato-Fishell-5 (Addgene, plasmid #83896), pAAV9-hDlx-GqDreadd-dTomoto-Fishell-4 (Addgene, plasmid # 83897), AAV-hDlx-dTomato (from Qian Chen) in Fig. 2 and Supplementary Fig. 2; pAAV8-Dlx-GCaMP6f-Fishell-2 (Addgene, plasmid #83899), AAV1-hSyn-HI-eGFP(nls) (from Qian Chen), AAV8-hSyn-DIO-hM3D(Gq)-mCherry (Addgene, plasmid # 44361) in Fig. 3 and Supplementary Fig. 4; pAAV8-hSyn-GRAB-eCB2.0 (Addgene, plasmid #164604) in Fig. 5 and Supplementary Fig. 10 were injected at bregma −1.31 mm, 2.15 mm to midline and 3.1−3.5 mm in depth. Immediately after virus injection, optical fiber (RWD Life Science, 200um core, 0.39 NA, OD 1.25 mm) was vertically implanted into the thalamus reticular nucleus in depth of 3.2 mm in Figs. 3, 5 and Supplementary Figs. 2, 3, 8, 10 and 11. Cannula fiber was fixed onto the mouse skull using adhesive luting cement (C&B-Metabond, #171032. Parkell. Edgewood, NY USA). All viruses were used at $0.5−1 × 10^{13}$ GC/ml.

For ventral posterior nuclei of thalamus injection, 20 nl of pENN.AAV9.CamKII.GCaMP6f.WPRE.SV40 (Addgene, Plasmid #100834) in Figs. 3, 6 and Supplementary Figs. 8, 9 and 14; AAV8-hSyn-DIO-hM4D(Gi)-mCherry (Addenge, plasmid # 44362), AAV1-hSyn-eGFP-nls (from Qian Chen) and pENN.AAV.hSyn.HI.eGFP-Cre.WPRE.SV40(Addgene, plasmid #105540) in Fig. 4 and S6 were injected at bregma − 1.55 mm, 1.7 mm to midline, 3.2 to 3.7 mm in depth. Optical fiber was implanted into the ventral posterior nuclei of thalamus in depth of 3.3 mm following virus injection. And adhesive luting cement was applied to fix the cannula fiber on the head skull. All viruses were used at $0.5−1 × 10^{13}$ GC/ml. For virus injection to the TRN and VP, behavioral testing was performed for the contralateral face and hindlimb.

## Fiber photometry
Fluorescent signals were recorded at desired time points using a multichannel fiber photometry system (Thinker Tech Nanjing Bioscience Inc, Nanjing China) as previously described[71–73]. Specifically,

optical fiber was prepared for imaging following virus injection 4 weeks ahead. For the fiber photometry imaging, the light beam of 473 nm LED (Cree XP-E LED) was reflected at a dichroic mirror (MD498, Thorlabs), the beam was focused through a ×20 objective lens (NA 0.4, Olympus) and then coupled to an optical commutator (Doric lenses). The individual optical fiber guided the light between the commutator and the implanted optical fiber in mice. Before starting imaging, the light intensity was set at to the low level of 0.01–0.02 mW at the tip of the optical fiber to minimize photobleaching. Fluorescent signal was generated by GCaMP excitation in mice and the signal was collected by the optical fiber. The GCaMP fluorescence signal was bandpass filtered (MF525-39, Thorlabs) and detected by the sensor of a CMOS (DCC3240M, Thorlabs) camera of the system. A Lab view program (Thinker Tech Nanjing Bioscience Inc, Nanjing China) was customized to control the CMOS camera and record calcium signal at a frequency of 50 HZ. Each mouse was imaged for 5–10 min at a time to minimize phototoxicity and bleaching.

### TRN cannulation and infusion
For TRN administration of NADA and behavioral examination, mice were anesthetized with isoflurane (3% for induction and 1.5% for maintenance) in oxygen. Mouse was placed onto a stereotactic frame and the head was positioned and fixed with two ear bars. Using an anesthetic musk to tighten the mouse snout over the incisor bar of the stereotactic frame, breathing rate was monitored during the procedure. The fur and skin on the top of mouse head was removed following skin preparation with Povidone-Iodine solution (Aplicare, INC., Neriden, CT USA) followed by 70% alcohol swab (BD, Franklin Lakes, USA). Using a scalp, the periosteum of the parietal skull was cleaned. Following craniotomy with a dental drill, a 26-gauge stainless steel guide cannula (C315GS, P1 Technologies, Roanoke, VA) was vertically inserted into the TRN at the coordinators of bregma −1.31 mm, 2.15 mm to midline and 3.2 mm in depth, the canula was covered by a dummy cannula (C315DCS, P1 Technologies). On the day of NADA administration and behavioral testing, the dummy cannula was removed and 33-gauge stainless steel internal cannula (C315IS, P1 Technologies) was inserted into the guide cannula. 20 nl of ACSF or NADA (0.6 mg/kg) dissolved in ACSF was slowly infused into the TRN. After that, mice were allowed to return to home cage and tested at desired time points.

TRN infusion for imaging of ventral posterior nuclei of thalamus: following virus injection and optical fiber implantation in the ventral posterior nuclei of thalamus, a guide cannula was inserted into TRN with a 30⁰ angle to vertical line at bregma−1.31 mm, 3.5 mm to midline and 2.6 mm in depth alone the angle. Both fiber ferrule and guide cannula were fixed on the dorsal skull with adhesive luting cement. On the day of experiment, between imaging sessions, either ACSF or NADA in ACSF (0.6 mg/kg) was slowly infused into the TRN. After TRN NADA administration, behavioral testing was performed for the contralateral face and hindlimb.

### Optogenetic manipulations of the TRN
TRN manipulations were facilitated by TRN microinjection of AAV-hSyn-eNpHR3.0-EYFP (Addgene 26972) or pAAV-mDlx-ChR2-mCherry (Addgene 83898). Optical cannulae were implanted into the TRN. Optogenetic manipulations were conducted using a IOS465 Intelligent Optogenetics System (RWD Life Sciences, San Diego, CA). For ChR2 experiments, wavelength of 465 nm was used. For eNpHR3.0 experiments, wavelength of 589 nm was used. Stimulation was conducted using 50 Hz frequency, pulse-width 2 ms (duty cycle: 10%) as described[36].

### Fluorescence/immunofluorescence imaging
Mouse was deeply anesthetized with 3–5% isoflurane for at least 5 minutes and perfused with 10 ml ice cold ×1 PBS followed by 30 ml 4% PFA, brain was extracted and immersed in 4% PFA at 4 °C

overnight. Then the brain was transfer into ×1 PBS for section. Using a vibratome (VT1000S, Leica), the brain was sliced with 50 μm in thickness. Slices covering bregma −0.5 mm to −2.0 mm were collected. Slices of mice brain injected virus was immediately used for fluorescence imaging using a confocal microscope (AXR, NIKON, Japan). For immunostaining, selected slices were blocked in 6% goat serum and 2% bovine serum albumin (BSA) in PBS with 0.2% Triton X-100 (Blocking solution) for one hour at room temperature followed by primary antibody (rabbit anti-VGLUT, MilliporeSigma, Catlog# G6642, 1:1000 dilution; rabbit anti-GABA, Sigma-Aldrich, Catlog# A2052,1:500 dilution; Guinea pig anti-TRPV1, Invitrogen, Catlog #PA129770, 1:500 dilution; rabbit anti-CB1 receptor, Invitrogen, Catlog# PA585080, 1:200 dilution; mouse anti-GFP polyclonal, Invitrogen, Catlog# A6455, 1:500 dilution; mouse anti-Parvalbumin, EMD Millipore Corporation, Catlog# MAB1572, 1:500 dilution; rabbit anti-c-Fos, Cell Signaling Technologies, Catlog#2250, 1:500 dilution) incubation in blocking solution at 4 °C for overnight. Slices were washed with 1× PBS 10 min for three times. Then the slices were incubated with second antibodies (Goat anti-rabbit Cy3, Jackson ImmunoResearch, Catlog# 111-165-003, 1:2000 dilution; Donkey anti-mouse FITC, Jackson ImmunoResearch, Catlog# 715-545-020, 1:1000 dilution; Goat anti-Guinea pig Cy3, Jackson ImmunoResearch, Catlog#: 106-165-003, 1:2000 dilution) at room temperature for one hour. Slices were mounted with DAPI (AbCam 104139) followed by image acquisition using a confocal microscope (AXR, NIKON, Japan). Images were later analyzed with ImageJ (NIH).

### Fiber photometry imaging analysis
The produced data by fiber photometry was processed and analyzed as described previously[73,74]. All data was analyzed using MATLAB (2018b, MathWorks, Cambridge, Unite Kingdom) embedded in Fiber Photometry machine (Thinker Tech Nanjing Bioscience Inc, Nanjing China) and 5 min of imaging data for each mouse was analyzed[73]. Changes of fluorescence intensity ($\Delta F/F_0$) was calculated as $\Delta F/F_0 = (V_{signal}-F_0)/F_0$, $F_0$ was defined the baseline fluorescence signal averaged over a 2.0-s-long control time window and $V_{signal}$ was the peak value of the fluorescence signal. And the values were normalized between 0 and 1 using the formular of $V_{signal} = (\Delta F/F-\min(\Delta F/F))/(\max(\Delta F/F)-\min(\Delta F/F))$, using a 50% cutoff, the total area under the curve (AUC) of $\Delta F/F_0$ was calculated for comparison.

### Gi and Gq DREADD activation for behavioral studies
C21 (Tocris, Catlog 5548, MN, 1 mg/kg) was intraperitoneally injected to the mice underwent DREADD/vector injection.

### Metabolomics
Mice subjected to chronic sleep deprivation underwent 5 consecutive days of sleep deprivation sessions. Sham mice were placed in the same room without any external interferences. At the end of the last session of sleep deprivation all mice were immediately sacrificed by decapitating using a disposable plastic cone with a guillotine for tissues collection. The cortex, thalamus and TRN were harvested respectively under an Omano surgical microscope (OM2300S-V7) at ×7–45 magnification. Tissues were frozen at −80 °C immediately for metabolomic analyzing. LC-MS/MS based metabolite profiling method.

A total of ~140 metabolites were measured in samples using MRM-based LC-MS metabolite profiling techniques as previously described[75,76]. Briefly, hydrophilic interaction liquid chromatography/positive ion mode MS detection to measure polar metabolites are conducted using an LC-MS system comprised of Agilent 1260 Infinity HPLC coupled to 4000-QTRAP mass spectrometer (Sciex). Samples (10 μL) were prepared via protein precipitation with the addition of nine volumes of 74.9:24.9:0.2 v/v/v acetonitrile/methanol/formic acid containing stable isotope-labeled internal standards (valine-d8, Sigma-Aldrich; St. Louis, MO; and phenylalanine-d8, Cambridge Isotope

Laboratories; Andover, MA). The samples were centrifuged (20 min, $15,000 \times g$, 4 °C), and the supernatants were injected directly onto a $150 \times 2$ mm, 3 μm Atlantis HILIC column (Waters). The column was eluted isocratically at a flow rate of 250 μL/min with 5% mobile phase A (10 mM ammonium formate and 0.1% formic acid in water) for 0.5 min followed by a linear gradient to 40% mobile phase B (acetonitrile with 0.1% formic acid) over 10 min. MS analyses were carried out using electrospray ionization in the positive ion mode using scheduled MRM method. Multiquant software (version 3.0.3, Sciex) was used for automatic peak integration followed by manual review of all peaks for quality of integration. Chemists were blinded to study sample assignments, using randomly generated tube numbers. Quality control samples were randomly inserted into sample sequence for quality assurance.

Central metabolites including sugars, sugar phosphates, organic acids, purine, and pyrimidines, were extracted from 30 μL of samples using acetonitrile and methanol and separated using a $100 \times 2.1$ mm 3.5-μm Xbridge amide column (Waters). Mobile phase A was 95:5 (v/v) water/acetonitrile, with 20 mM ammonium acetate and 20 mM ammonium hydroxide (pH 9.5). Mobile phase B was acetonitrile. Tandem MS analysis for negative mode detection utilizes a high sensitivity Agilent 6490 QQQ mass spectrometer equipped with an electrospray ionization source. The settings were as follows: sheath gas temperature, 400 °C; sheath gas flow, 12 l/min; drying gas temperature, 290 °C; drying gas flow, 15 l/min; capillary, 4,000 V; nozzle pressure, 30 psi; nozzle voltage, 500 V; and delta EMV, 200 V. Detailed methods have been described previously[75,76]. Raw data were processed using MassHunter Quantitative Analysis Software (Agilent). Volcano plots were generated using Python Seaborn package.

### Reporting summary

Further information on research design is available in the Nature Portfolio Reporting Summary linked to this article.

## Data availability

The construct of AAV-hSyn-GFP (nls) will be deposited to Addgene for access. Original imaging is available upon request. Source data are provided in this paper.

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

## Acknowledgements

The authors acknowledge MGH IACUC and the animal facility for their kind support; Scot Mackeil from MGH Bioengineer Lab for anesthesia equipment maintenance and validation; Department of Anesthesia, Critical Care and Pain Medicine of MGH for generous support. The authors thank Dr. Tong Zhu for critical discussion, and Shelley Turok, David Duarte, and Ariel Mueller for administrative support. Shiqian Shen lab received support from NIH R61NS116423, NIH R35GM128692, NIH R01 AG 070141, NIH R03 AG067947, and NIH R61 NS126029. Part of this work was supported by NSF EAGER 2334666. Weihua Ding received support from the Borsook Project. The K. Lisa Yang and Hock E. Tan Center for Molecular Therapeutics in Neuroscience at MIT supported this work in Guoping Feng lab.

## Author contributions

Conceptualization: G.F., S.S.; methodology: W.H.D., Q.C., G.F., S.S.; investigation: W.H.D., L.Y., Q.C.; E.S., B.K., S.L., K.H., L.G., P.C., We.D., D.B., A.L., J.C., C.W., J.Y., C.R., K.S., O.A., J.M.; writing original draft: S.S., W.H.D; editing of the manuscript: W.H.D., G.F.; funding acquisition: S.S.

## Competing interests

The authors declare no competing interests.
