## [Peer Review File · Nature Communications]

Reviewers' Comments:

Reviewer #1:

Remarks to the Author:

Ding et al. examined the mechanisms of pain hypersensitivity induced by Chronic sleep disorders (CSD). They report the role of the endocannabinoid, N-arachidonoyl dopamine (NADA) signaling in CSD-induced hyperalgesia. The study identified the importance of the thalamic reticular nucleus, a brain region involved in sleep and arousal in CSD-induced hyperalgesia. They show that TRN projects to the ventral posterior nucleus of the thalamus. Chemogenetic activation of TRN neurons or inhibition of VP neurons, both attenuated CSD-induced hyperalgesia. Moreover, they found that CSD reduces the level of NADA in the TRN. In contrast, NADA administration to the TRN attenuates CSD-induced hyperalgesia.

This is an interesting study that provides novel and important information on the mechanisms of chronic pain associated with sleep disorders. Using chemogenetics combined to behavioural analysis, the study confirms the role of the TRN in CSD-induced hyperalgesia. The conclusion that NADA acts on TRN neurons is supported by in vivo Fiber photometry.

I think that a couple of experiments are missing to support the conclusion of the paper:

- In Figure 2, Fiber photometry recordings following chemogenetics inhibition should be presented to support the behavioral changes. Showing that activity in the TRN is altered upon C21 administration would strengthen the conclusion that DREADD-induced modulation of CSD-induced hyperalgesia occurs through inhibition of TRN neurons.
- While a c-Fos mapping was not performed in response to CSD, the increase in neural activity shown in Fig. 3e and f would be more convincing if c-Fos activation was detected in VP neurons after CSD.
- The authors show that NADA level decreases in CSD and NADA infusion alleviates CSD induced hyperalgesia. However, the mechanisms of action of NADA is unclear. Is the behavioral effect of NADA mediated by CB1 receptor? Does a CB1 antagonist prevent the relief of hyperalgesia following NADA administration? I think these experiments are important to address how NADA modulates pain sensitivity in the context of sleep deprivation. Moreover, proof of concept experiment using a CB1 agonist should be included to validate the functional expression of the GRABeCB2.0 in the TRN.

Minor concerns:

The immunofluorescence imaging should include DAPI as a nuclear marker, particularly in Fig. S3c where the AAV1-hSyn-eGFP(nls) construct is expected to localize in the nucleus. Higher magnification images should be presented.

Region of interest presented in Fig. S3i should be indicated in 3h, as for S3f and S3e.

How many mice were used in Fig. S2

L171: TRN neurons there were "that were"

L241: Mechanistically, this role of TRN is likely mediate "mediated" by its projections to the VP.

In the keywords, Endocannabinoid receptor 1 should be replaced by cannabinoid receptor 1

Reviewer #2:

Remarks to the Author:

In their article entitled "The endocannabinoid N-arachidonoyl dopamine is critical for hyperalgesia induced by chronic sleep disruption", the authors test the role of NADA on sleep loss-induced hyperalgesia in mice. First, they show how 5d of partial sleep deprivation cause an increase in pain sensitivity in the facial and hindpaw territories. Using viral injections of GCaMP construct, they next show that sleep loss causes a decrease in activity in the reticular thalamic nucleus (TRN) GABAergic neurons and an increase of activity of ventralbasal thalamus (VP) glutamatergic neurons contacted by TRN. Chemogenetic activation of TRN neurons causes an inhibition of VP neurons contacted. In well-rested mice, chemogenetic inhibition of TRN neurons increases pain responses, while chemogenetic activation restores pain sensitivity in sleep-deprived mice. In a next step of experiments, the authors show that chemogenetic inhibition of VP neurons restores pain sensitivity in sleep deprived mice. Altogether, these results indicate an involvement of the TRN-VP pathway in the increased pain sensitivity caused by sleep loss. Next, the authors find that sleep loss causes a downregulation in NADA levels in TRN and a loss of function of CB1 receptors expressed by GABA TRN neurons projecting to VP. Infusion of NADA in TRN restores pain sensitivity in sleep-deprived mice and this is associated with a decrease in TRN neurons.

Overall this is an extremely nice study, very pleasant to read and that convincingly show a role of TRN-VP pathway in sleep loss-induced hyperalgesia. A few controls are missing, especially since several viruses used rely on their promoter only. Some stainings to confirm correct expression profile are important to include.

The effects of inhibition of TRN neurons on pain should be assessed in well-rested mice.

Additional behavior readouts should be added in TRN inhibition experiments, including some general behavioral assays to confirm the effects are specific to pain responses.

Did the authors assess the effects of activation or inhibition of TRN neurons on sleep?

NADA experiments. I have a major problem here. The authors show clearly how sleep loss reduces activity of TRN neurons, increases VP activity. They next show NADA levels drop after CSD and this is associated with reduced CB1 activity. CB1 is expressed in PV (hence GABA) TRN neurons. CB1 causes inhibition of neurons, so in a simplistic manner, CSD that reduces NADA-CB1 should cause a gain in activity (loss of normal inhibition) of TRN neurons. Yet sleep loss causes a decrease in activity of GABAergic TRN neurons. The loss of activity of TRN GABA neurons after sleep loss cannot be caused by a loss of CB1 signaling, unless CB1 changes its coupling? I might have missed something, but it seems the results with a loss of NADA goes against the TRN-VP circuit proposed.

Regarding GRAB vector - the authors mention GCaMP as well. Is that a typo? Isn't the construct based on GFP signal?

Reviewer #3:

Remarks to the Author:

Chronic sleep disruption (CSD) is a well known factor that contributes to the worsening of pain conditions and is likely associated with alterations in thalamocortical signaling. The underlying mechanisms are unclear. The study "The endocannabinoid N-arachidonoyl dopamine is critical for hyperalgesia induced by chronic sleep disruption" by Ding et al. proposes that following CSD, N-arachidonoyl dopamine (NADA) levels are decreased in TRN, which leads to decreased TRN baseline activity. In turn, VP becomes hyperactive, due to the lack of TRN mediated inhibition and thereby presumably transfers nociceptive signals more efficiently to the cortex. The results of CSD mediated changes in thalamocortical signaling are highly relevant and make important contributions to the field. The paper is well written and to the point. However, there are some points that should be addressed.

1. How specific is this proposed mechanism for CSD? Would similar results, in particular the reduction

of NADA in the TRN and VP hyperactivity be expected in other pain models?

2. The relationship of the TRN to pain/nociception should be acknowledged, there are studies describing a role of the TRN in pain and the underlying circuit mechanisms which were partly reproduced here. For example, reduced GABAergic transmission from TRN to the VP contributes to increased nociception and activation of the TRN-VP pathway relieves pain e.g. thermal hyperalgesia in chronic inflammatory pain (e.g. DOI:10.1038/srep41439, <https://doi.org/10.1016/j.brainres.2022.148174>).

3. Fig. 2: Chemogenetic inhibition of the TRN promoted nociceptive behavior. I could not find data showing that TRN is inhibited and thus this conclusion cannot be drawn based on the data that is shown. Gi is not always inhibiting neurons. Furthermore, TRN hyperpolarization can lead to increased spike output (the opposite of what is presumed in these experiments), due to the deinactivation of T-type Ca-channels. Functional characterization of what Gi is doing to TRN output is necessary to validate this tool and and/or you could validate the above conclusion with an alternative approach (e.g. optogenetic inhibition). The same applies to the DREADD experiments in Fig. 4, which lack functional characterization of what the DREADDs (Gi and Gq) are doing to thalamic activity.

4. Bidirectional connectivity between TRN and VP (Figure S3): I did not understand how the anterograde and retrograde approaches can be used to investigate the connectivity between TRN and VP as both, the starter and the recipient neurons, express the same marker. In this setting discrimination between these two populations is not possible and I don't see how connectivity can be inferred from the data that is shown. For example, the anterograde injection in S3c could have been accidentally placed into the VP and would show a very similar pattern (VB projects to TRN, so both populations would be labeled). Furthermore, the investigated TRN region also projects to PO and thus PO should contain labeled neurons (and should also be affected by the DREADD and pharmacological manipulations that were done in TRN). Can you confirm labeling of PO in the anterograde experiment? Since the bidirectional connectivity between TRN and VB is well established, the authors could remove these experiments and instead refer to the literature. If they want to reconfirm this point, I suggest using an approach in which the starter neurons and recipient neurons have different markers or some other intersectional strategy.

5. The authors should clarify which thalamic system they investigate. VP thalamus is mentioned throughout the study but this is rather unspecific. Based on the connectivity data in Figs. S3 and S4, it seems the study was conducted in the face/whisker thalamus (VPM). It should be clarified how this relates to the behavioral tests, in which the hind paws were stimulated, which are represented not in VPM but in the VPL and its associated TRN region but Fig. S3 shows very little labeling in VPL.

6. The role of POM is not considered, even though it is also bidirectionally connected with the TRN and has been repeatedly implicated in pain. In principle, the idea presented here, that TRN activity is decreased as a result of CSD, which then leads to increased thalamocortical throughput of sensory signals could be happening in the TRN-POM pathway as well.

7. The study finds increased baseline activity in VP following CSD. This is an important finding, however, since the study is built on evoked-pain models (face & hindpaw stimulation), the question whether such stimulation leads to increased VP activity remains open. In order to establish the intended link between CSD-induced hypersensitivity and putative increased thalamocortical throughput of sensory signals, the authors would need to test if sensory-evoked responses in VP are indeed affected by CSD and TRN manipulations.

8. To activate DREADDs the study uses C21, which has been described to have a small window of selectivity to activate hM4Di in rodents. Can the authors comment on their choice of C21, in comparison to J60?

Minor

line 103: ...in mice that underwent CSD

line 117: reference to figure 1b seems incorrect

Fig.2a: scale bars are labeled in the first and last panel, but not in the others. I assume the magnification is the same in all panels and would thus suggest to remove the label in the last panel

line 119: "an excitatory DREADDs AAV-Dlx-Gq DREADDs were" → "an excitatory DREADD AAV-Dlx-Gq DREADDs was"

line 133: remove "own" or reformulate

line 169: were → was

line 180: Mice underwent → Mice that underwent

Reviewer #1 (Remarks to the Author):

Ding et al. examined the mechanisms of pain hypersensitivity induced by Chronic sleep disorders (CSD). They report the role of the endocannabinoid, N-arachidonoyl dopamine (NADA) signaling in CSD-induced hyperalgesia. The study identified the importance of the thalamic reticular nucleus, a brain region involved in sleep and arousal in CSD-induced hyperalgesia. They show that TRN projects to the ventral posterior nucleus of the thalamus. Chemogenetic activation of TRN neurons or inhibition of VP neurons, both attenuated CSD-induced hyperalgesia. Moreover, they found that CSD reduces the level of NADA in the TRN. In contrast, NADA administration to the TRN attenuates CSD-induced hyperalgesia.

This is an interesting study that provides novel and important information on the mechanisms of chronic pain associated with sleep disorders. Using chemogenetics combined to behavioural analysis, the study confirms the role of the TRN in CSD-induced hyperalgesia. The conclusion that NADA acts on TRN neurons is supported by in vivo Fiber photometry.

I think that a couple of experiments are missing to support the conclusion of the paper:

- In Figure 2, Fiber photometry recordings following chemogenetics inhibition should be presented to support the behavioral changes. Showing that activity in the TRN is altered upon C21 administration would strengthen the conclusion that DREADD-induced modulation of CSD-induced hyperalgesia occurs through inhibition of TRN neurons.

We've conducted experiments as suggested: we injected a 1:1 mixture of AAV-Dlx-Gi DREADDs-dTomato and AAV-Dlx-GCaMP6f to the TRN. Mice were allowed for 4 weeks for viral expression. Calcium imaging was performed prior to and 30 minutes after C21 administration (1mg/kg, i.p.). TRN neuronal activities significantly decreased after C21 administration. We have now included these data in our revised manuscript **Figure S2b and c**.

- While a c-Fos mapping was not performed in response to CSD, the increase in neural activity shown in Fig. 3e and f would be more convincing if c-Fos activation was detected in VP neurons after CSD.

We have now conducted c-Fos staining in the thalamus and observed increased c-Fos expression in VP neurons following CSD (**Figure S7a to c**). These data were consistent with previous reports that noxious mechanical and thermal stimuli increased c-Fos expression in the VP (PMID: 2113539).

- The authors show that NADA level decreases in CSD and NADA infusion alleviates CSD induced hyperalgesia. However, the mechanisms of action of NADA is unclear. Is the behavioral effect of NADA mediated by CB1 receptor? Does a CB1 antagonist prevent the relief of hyperalgesia following NADA administration? I think these experiments are important to address how NADA modulates pain sensitivity in the context of sleep deprivation. Moreover, proof of concept experiment using a CB1 agonist should be included to validate the functional expression of the GRABeCB2.0 in the TRN.

We have conducted suggested experiments. Using a CB1 receptor antagonist SR141716A (Rimonabant, 10mg/kg, i.p.), the relief of CSD-induced hyperalgesia by NADA was abrogated, suggesting CB1 receptor was implicated in NADA's effects. These data have been included as **Figure S13a to d**.

Additionally, we have also conducted experiments using a CB1 agonist Arachidonyl-2'-chloroethylamide (ACEA, 7.5mg/Kg, i.p.) and observed significantly increased GRAB-eCB2.0 activities using fiber photometry, validating the TRN imaging using GRAB-eCB2.0. These data have been added as **Figure S11a and b**.

Minor concerns:

The immunofluorescence imaging should include DAPI as a nuclear marker, particularly in Fig. S3c where the AAV1-hSyn-eGFP(nls) construct is expected to localize in the nucleus. Higher magnification images should be presented.

We have now revised previous **Figure S3 to Figure S4** and included DAPI as a nuclear marker as suggested, as well as included higher magnification images.

Region of interest presented in Fig. S3i should be indicated in 3h, as for S3f and S3e.

We have made corrections accordingly.

How many mice were used in Fig. S2

N=8 each group for behavior assessment was added in Figure S2 legend.

L171: TRN neurons there were “that were”

We have made the suggested correction.

L241: Mechanistically, this role of TRN is likely mediate “mediated” by its projections to the VP.

We have made the suggested correction.

In the keywords, Endocannabinoid receptor 1 should be replaced by cannabinoid receptor 1

We have made the suggested correction.

Reviewer #2 (Remarks to the Author):

In their article entitled “The endocannabinoid N-arachidonoyl dopamine is critical for hyperalgesia induced by chronic sleep disruption”, the authors test the role of NADA on sleep loss-induced hyperalgesia in mice. First, they show how 5d of partial sleep deprivation cause an increase in pain sensitivity in the facial and hindpaw territories. Using viral injections of GCaMP construct, they next show that sleep loss causes a decrease in activity in the reticular thalamic

nucleus (TRN) GABAergic neurons and an increase of activity of ventralbasal thalamus (VP) glutamatergic neurons contacted by TRN. Chemogenetic activation of TRN neurons causes an inhibition of VP neurons contacted. In well-rested mice, chemogenetic inhibition of TRN neurons increases pain responses, while chemogenetic activation restores pain sensitivity in sleep-deprived mice. In a next step of experiments, the authors show that chemogenetic inhibition of VP neurons restores pain sensitivity in sleep deprived mice. Altogether, these results indicate an involvement of the TRN-VP pathway in the increased pain sensitivity caused by sleep loss. Next, the authors find that sleep loss causes a downregulation in NADA levels in TRN and a loss of function of CB1 receptors expressed by GABA TRN neurons projecting to VP. Infusion of NADA in TRN restores pain sensitivity in sleep-deprived mice and this is associated with a decrease in TRN neurons.

Overall this is an extremely nice study, very pleasant to read and that convincingly show a role of TRN-VP pathway in sleep loss-induced hyperalgesia. A few controls are missing, especially since several viruses used rely on their promoter only. Some stainings to confirm correct expression profile are important to include.

DAPI for nuclear staining has been added (Revised Figure S4). In the original and revised manuscript, we included staining for GABA and glutamate to the TRN-VP pathway to demonstrate the correction expression profile (Figure S4 and 6).

The effects of inhibition of TRN neurons on pain should be assessed in well-rested mice. Additional behavior readouts should be added in TRN inhibition experiments, including some general behavioral assays to confirm the effects are specific to pain responses. Did the authors assess the effects of activation or inhibition of TRN neurons on sleep?

The TRN inhibition experiment was conducted in naive animals that were well-rested, this has now been clarified in the revised manuscript. Besides mechanical and thermal assays, we have now performed an open field test, to show that TRN inhibition did not induce anxiety-like behavior, or significantly alter locomotion. As such, the observed hyperalgesia behavior was likely related to altered pain-processing, but not anxiety or locomotive issues.

Reviewer perusal figure 1: TRN inhibition does lead to anxiety-like behavior in well-rested C57/BL6 mice. a) Sketch depicts open field test; b) Time spent in center area and b) Number of

entrances to the center area; d) Total distance traveled. N=8, two-tailed unpaired t test, NS: $p > 0.05$.

We did not conduct experiments on TRN modulation and sleep, as this has been investigated by other groups previously. For example, Lewis et al showed that TRN activation induces slow waves activity in a spatially restricted region of the cortex. These slow waves resemble those seen in sleep (PMID 26460547). Halassa *et al* showed that brief selective drive of TRN could switch thalamocortical firing mode from tonic to bursting and generate neocortical sleep spindles, which was confirmed by Thankachan *et al*. (PMID 21785436; 30837664).

NADA experiments. I have a major problem here. The authors show clearly how sleep loss reduces activity of TRN neurons, increases VP activity. They next show NADA levels drop after CSD and this is associated with reduced CB1 activity. CB1 is expressed in PV (hence GABA) TRN neurons.

CB1 causes inhibition of neurons, so in a simplistic manner, CSD that reduces NADA-CB1 should cause a gain in activity (loss of normal inhibition) of TRN neurons. Yet sleep loss causes a decrease in activity of GABAergic TRN neurons. The loss of activity of TRN GABA neurons after sleep loss cannot be caused by a loss of CB1 signaling, unless CB1 changes its coupling? I might have missed something, but it seems the results with a loss of NADA goes against the TRN-VP circuit proposed.

Our fiber photometry results showed that the TRN activities were decreased by CSD, which was accompanied by decreased levels of TRN NADA levels. As the Reviewer pointed out, CB1 receptor located at the presynaptic terminals negatively regulates neuronal activities through retrograde signaling, which implies that decreased NADA levels would enhance TRN activities. It is plausible that non-retrograde endocannabinoid signaling pathways were implicated in our observations. For example, CB1 receptor has been shown to localize in mitochondrial membrane (PMID: 32641832, *Nature* 2020; PMID 25707796, *Nature* 2015; and PMID 22388959, *Nature Neuroscience* 2012), which plays fundamental roles in neuronal energy metabolism and host behaviors. For example, CB1R has been shown to protect mitochondria function and alleviate oxidative stress (PMID 26215450;33536881;30091204). In line with these reports, our TRN metabolomics data showed that upregulated metabolites after CSD include **glycerol-3 phosphate and glutathione (Figure 5a)**, both of which are key molecules implicated in energy metabolism and oxidative stress. In an additional experiment, we directly assessed reactive oxygen species using an oxalate-curcumin-based probe (PMID 29109280) with IVIS imaging. Results showed that CSD led to significantly higher levels of TRN reactive oxygen species than control. Previously, the production of reactive oxygen species has been shown to increase following sleep loss in both flies and mice (PMID: 32502393).

PMID: 29109280. *** $p < 0.001$, t test.

Regarding GRAB vector - the authors mention GCaMP as well. Is that a typo? Isn't the construct based on GFP signal?

That was a typo. The construct was based on GFP signals as described in PMID 34764491.

Reviewer #3 (Remarks to the Author):

Chronic sleep disruption (CSD) is a well-known factor that contributes to the worsening of pain conditions and is likely associated with alterations in thalamocortical signaling. The underlying mechanisms are unclear. The study “The endocannabinoid N-arachidonoyl dopamine is critical for hyperalgesia induced by chronic sleep disruption” by Ding et al. proposes that following CSD, N-arachidonoyl dopamine (NADA) levels are decreased in TRN, which leads to decreased TRN baseline activity. In turn, VP becomes hyperactive, due to the lack of TRN mediated inhibition and thereby presumably transfers nociceptive signals more efficiently to the cortex. The results of CSD mediated changes in thalamocortical signaling are highly relevant and make important contributions to the field. The paper is well written and to the point. However, there are some points that should be addressed.

1. How specific is this proposed mechanism for CSD? Would similar results, in particular the reduction of NADA in the TRN and VP hyperactivity be expected in other pain models?

We have conducted an additional experiment and examined NADA in a surgical incision pain model as described in PMID 8783314. Results showed that NADA was not significantly altered in this model, suggesting differential pain mechanisms implicated in different pain models.

Reviewer perusal figure 3. Mice underwent either sham or plantar foot incision (N=8/group). TRN samples were obtained at day 5 post surgery which were subsequently quantified for NADA concentrations by mass spectrometry. **a)** Experimental design. **b)** NADA quantification. NS, $p > 0.05$ t test.

2. The relationship of the TRN to pain/nociception should be acknowledged, there are studies describing a role of the TRN in pain and the underlying circuit mechanisms which were partly reproduced here. For example, reduced GABAergic transmission from TRN to the VP contributes

to increased nociception and activation of the TRN-VP pathway relieves pain e.g. thermal hyperalgesia in chronic inflammatory pain (e.g. DOI:10.1038/srep41439, <https://doi.org/10.1016/j.brainres.2022.148174>).

We have added relevant references as suggested, including PMID 29042322; 28150719; 36427592.

3. Fig. 2: Chemogenetic inhibition of the TRN promoted nociceptive behavior. I could not find data showing that TRN is inhibited and thus this conclusion cannot be drawn based on the data that is shown. Gi is not always inhibiting neurons. Furthermore, TRN hyperpolarization can lead to increased spike output (the opposite of what is presumed in these experiments), due to the deinactivation of T-type Ca-channels. Functional characterization of what Gi is doing to TRN output is necessary to validate this tool and and/or you could validate the above conclusion with an alternative approach (e.g. optogenetic inhibition). The same applies to the DREADD experiments in Fig. 4, which lack functional characterization of what the DREADDs (Gi and Gq) are doing to thalamic activity.

As suggested by both this Reviewer and Reviewer 1, we have conducted two experiments to validate the Gi DREADDs strategy that were used for Figure 2 and Figure S2. Specifically, we performed optogenetic inhibition of the TRN (**Figure S3**) and characterized VP thalamic activities following TRN manipulation (**Figure S8**).

In revised **Figure S2**, we injected a 1:1 mixture of AAV-Dlx-Gi DREADDs-dTomato and AAV-Dlx-GCaMP6f to the TRN. Mice were allowed for 4 weeks for viral expression. Calcium imaging was performed prior to and 30 minutes after C21 administration. TRN neuronal activities as measured by calcium signals significantly decreased after C21 administration, suggesting the Gi DREADDs led to decreased TRN neuronal activities.

Optogenetic inhibition of the TRN leads to hyperalgesia. AAV-hSyn-eNpHR-EYFP (Addgene 26972) or AAV-hSyn-YFP (a gift from Dr. Qian Chen) was injected into the TRN (N=8), which was followed by fiber implantation. Halorhodopsin-mediated TRN inhibition led to hyperalgesia behavior similar to that observed with Gi DREADDs (**Figure S3a to d**).

Optogenetic activation of the TRN. AAV-mDlx-ChR2-mCherry (Addgene 83898) or vector virus was injected into the TRN; AAV-CaMKII-GCaMP6f was injected into the VP (N=6/group). Following baseline imaging of VP activity and behaviors test, mice received 5 sessions of CSD. Optogenetic activation was conducted at the end of the final CSD session accompanied by VP imaging. In CSD animals that received ChR2, optogenetic manipulation led to decreased VP activities (**Figure S8**). Additionally, optogenetic activation of the TRN alleviated hyperalgesia behavior in CSD mice (**Figure S3e to h**).

4. Bidirectional connectivity between TRN and VP (**Figure S3**): I did not understand how the anterograde and retrograde approaches can be used to investigate the connectivity between TRN and VP as both, the starter and the recipient neurons, express the same marker. In this setting discrimination between these two populations is not possible and I don't see how connectivity can be inferred from the data that is shown. For example, the anterograde injection in S3c could have been accidentally placed into the VP and would show a very similar pattern

(VB projects to TRN, so both populations would be labeled). Furthermore, the investigated TRN region also projects to PO and thus PO should contain labeled neurons (and should also be affected by the DREADD and pharmacological manipulations that were done in TRN). Can you confirm labeling of PO in the anterograde experiment? Since the bidirectional connectivity between TRN and VB is well established, the authors could remove these experiments and instead refer to the literature. If they want to reconfirm this point, I suggest using an approach in which the starter neurons and recipient neurons have different markers or some other intersectional strategy.

Following a method described by Martinez-Garcia RI et al.(PMID: 32699410) , a Cre-dependent AAV-hSyn-DIO-GFP was injected to the TRN of PV-Cre mice. Previously, it has been shown that TRN is highly enriched for PV expressing neurons, and that there were no significant PV signals expressed in mouse thalamus PMID: 32699410. Using this strategy, we confirmed robust projections from the TRN to the VP. Additionally, we also observed TRN projections to other thalamic regions, including laterodorsal dorsomedial thalamus (LDDM), ventral lateral part (LDVL)and the PO region (Figure S5).

5. The authors should clarify which thalamic system they investigate. VP thalamus is mentioned throughout the study but this is rather unspecific. Based on the connectivity data in Figs. S3 and S4, it seems the study was conducted in the face/whisker thalamus (VPM). It should be clarified how this relates to the behavioral tests, in which the hind paws were stimulated, which are represented not in VPM but in the VPL and its associated TRN region but Fig. S3 shows very little labeling in VPL.

For the neural circuitry dissection in Figure S6, we intentionally targeted VPM instead of both VPM and VPL by sparing an anatomical gap (VPL between VPM and TRN), to better demonstrate the projections between TRN and VP, following virus injection strategy as described in (PMID 32699410). For VP imaging, both VPM and VPL were targeted with viruses as demonstrated in the Reviewer perusal figure 4. a) Sketch depicts virus injecting into VP covering both VPM and VPL. b) A representative image of brain slices validating virus expression covering both VPM and VPL. N=3.

6. The role of POM is not considered, even though it is also bidirectionally connected with the TRN and has been repeatedly implicated in pain. In principle, the idea presented here, that TRN activity is decreased as a result of CSD, which then leads to increased thalamocortical throughput of sensory signals could be happening in the TRN-POM pathway as well.

As the Reviewer pointed out, the functional connection between the POM and TRN has been well characterized. However, our c-Fos staining results showed neural activity was remarkably

activated in the VP (Figure S7) but not in the POm after CSD. As such, we focused our investigation on the TRN-VP pathway.

Reviewer perusal figure 5. Mice received 5 sessions of CSD or Sham (N=6/group) and were sacrificed by the end of the last session, slices between bregma -1.07 and -1.55 mm were collected for c-Fos staining, VP region was identified according to the mouse brain atlas which was confirmed by its proximity with TRN (stained with anti-PV marker). a) Representative brain slice demonstrates the location of PO. b) Number of c-Fos+ cells. Unpaired t test; NS: $p > 0.05$.

7. The study finds increased baseline activity in VP following CSD. This is an important finding, however, since the study is built on evoked-pain models (face & hindpaw stimulation), the question whether such stimulation leads to increased VP activity remains open. In order to establish the intended link between CSD-induced hypersensitivity and putative increased thalamocortical throughput of sensory signals, the authors would need to test if sensory-evoked responses in VP are indeed affected by CSD and TRN manipulations.

We have conducted experiments as suggested by the reviewer and added to revised manuscript (Figure S9). We found that VP reactivity to mechanical stimulation was enhanced after CSD sessions, and that chemogenetic activation of the TRN could ameliorate CSD-induced VP hyperreactivities.

8. To activate DREADDs the study uses C21, which has been described to have a small window of selectivity to activate hM4Di in rodents. Can the authors comment on their choice of C21, in comparison to J60?

The prototypical DREADDs agonist CNO, has been shown to have limited brain penetration and biologically active metabolites. As such, newer generation of DREADDs agonists such as C21, J60 etc. have been developed. We chose C21 based on our own experience with this drug (PMID 36602876) and other reports (PMID 37141366). However, we agree that J60 would be an excellent alternative.

Minor:

line 103: ...in mice that underwent CSD ...

line 117: reference to figure 1b seems incorrect

Fig.2a: scale bars are labeled in the first and last panel, but not in the others. I assume the magnification is the same in all panels and would thus suggest to remove the label in the last panel

line 119: “an excitatory DREADDs AAV-Dlx-Gq DREADDs were” → “an excitatory DREADD AAV-Dlx-Gq DREADDs was”

line 133: remove “own” or reformulate

line 169: were → was

line 180: Mice underwent → Mice that underwent

We have made all suggested changes as listed above.

REVIEWERS' COMMENTS

Reviewer #1 (Remarks to the Author):

Ding et al. provided additional experiments and addressed all of my comments. I only have minor corrections:

- 1) In Figure S3e, please correct "mCharry" to "mCherry" on the figure.
- 2) In Figure S7, please clarify whether the number of cFos+ cells is reported per mouse unit or brain sections.

Reviewer #2 (Remarks to the Author):

The authors have performed a tremendous number of additional experiments to answer all queries, and replied thoroughly to all questions.

The revised manuscript is extremely clear and well presented, and provides a major mechanistic insight into sleep loss-induced hyperalgesia.

The CB1R results are still a little bit problematic and I think this will require more work in the future to be able to link formally CB1R activity in mitochondria of GABAergic TRN neurons to the changes in activity of these neurons after sleep loss. However, the authors do an excellent job at presenting and discussing the results in the current MS.

Congratulations on this great study.

Reviewer #3 (Remarks to the Author):

The authors have addressed all of the reviewers points and provide several additional experiments that strengthen their study. The revised manuscript is well-written, and I would like to congratulate the authors. From my side, there is one important remaining point:

The new experiment using AAV-hSyn-DIO-GFP in PV-Cre mice (Figure S5), needs more explanation. Why are the GFP signals in VPL/VPM/PO enriched in the somata? This can only be understood if the virus has trans-synaptic transduction properties, such that it either jumps from TRN to the postsynaptic neurons or the virus is retrogradely taken up by presynaptic neurons (the anterograde scenario is consistent with the images in S5, in case of a retrograde virus, one would also expect somata in S1-layer 6, which is not the case). Otherwise, if this was a non-jumping virus (which I thought was the aim of the experiment), one would expect axonal signals, rather than somatic signals (TRN axons projecting to thalamic relay neurons), which is not what can be seen in S5. This should be clarified.

Reviewer #1 (Remarks to the Author):

Ding et al. provided additional experiments and addressed all of my comments. I only have minor corrections:

- 1) In Figure S3e, please correct "mCharry" to "mCherry" on the figure.
- 2) In Figure S7, please clarify whether the number of cFos+ cells is reported per mouse unit or brain sections.

We have made the suggested changes.

Reviewer #2 (Remarks to the Author):

The authors have performed a tremendous number of additional experiments to answer all queries, and replied thoroughly to all questions.

The revised manuscript is extremely clear and well presented, and provides a major mechanistic insight into sleep loss-induced hyperalgesia.

The CB1R results are still a little bit problematic and I think this will require more work in the future to be able to link formally CB1R activity in mitochondria of GABAergic TRN neurons to the changes in activity of these neurons after sleep loss. However, the authors do an

excellent job at presenting and discussing the results in the current MS.
Congratulations on this great study.

Thank you.

Reviewer #3 (Remarks to the Author):

The authors have addressed all of the reviewers points and provide several additional experiments that strengthen their study. The revised manuscript is well-written, and I would like to congratulate the authors. From my side, there is one important remaining point:

The new experiment using AAV-hSyn-DIO-GFP in PV-Cre mice (Figure S5), needs more explanation. Why are the GFP signals in VPL/VPM/PO enriched in the somata? This can only be understood if the virus has trans-synaptic transduction properties, such that it either jumps from TRN to the postsynaptic neurons or the virus is retrogradely taken up by presynaptic neurons (the anterograde scenario is consistent with the images in S5, in case of a retrograde virus, one would also expect somata in S1-layer 6, which is not the case). Otherwise, if this was a non-jumping virus (which I thought was the aim of the experiment), one would expect axonal signals, rather than somatic signals (TRN axons projecting to thalamic relay neurons), which is not what can be seen in S5. This should be clarified.

Thank you for the comment. As Reviewer 3 pointed out, AAV-hSyn-DIO-GFP in PV-Cre mice (Figure S5) experiment was conducted using AAV1, a serotype with known transsynaptic properties (PMID: 27989459). We have now added this clarification in the figure legends.

We greatly appreciate your time and effort in helping us improving our manuscript.